# Microglial colonization of the developing mouse brain is controlled by both microglial and neural CSF-1

Cécile Bridlance[1,2,3], Sarah Viguier[1,2,3], Nicolas Olivié[1,2,3], Edmond Dupont [1], Dorine Thobois[1], Benjamin Mathieu[2], Jean X Jiang [4], Guillermina López-Bendito[5], Melanie Greter[6], Burkhard Becher[6], Florent Ginhoux [7,8], Aymeric Silvin[7], Esther Klingler[9,10], Sonia Garel [1,2,11,12 ✉] & Morgane Sonia Thion [1,2,12 ✉]

## Abstract

**Microglia are brain-resident macrophages critical for cerebral development, function, and homeostasis. During development, yolk sac-derived microglial progenitor cells colonize and populate the brain following a well-defined spatiotemporal pattern. However, the mechanisms controlling microglial colonization and proliferation remain largely unknown. Here, we describe two broad waves of microglial proliferation in the developing mouse forebrain. Microglia accumulate in transient hotspots, in a proliferative axon tract-associated microglia (ATM)-like state. Prenatal and early postnatal patterns of microglial colonization do not rely on neuronal activity. Instead, using conditional inactivation of the microglial regulator colony-stimulating factor 1 (*Csf1*) gene, we reveal that the distribution and proliferation of embryonic cortical microglia critically rely on neural CSF-1, mainly produced by cortical progenitor cells but also by post-mitotic neurons, with the action of CSF-1 being local, dose-dependent, and transient. In addition, intrinsic CSF-1 expressed by ATM microglia contributes to their sustained proliferation in developmental hotspots. Our study reveals that microglia rely on distinct, local, and cell-type-specific sources of CSF-1 for their developmental distribution, which has major implications for understanding how microglia colonize the brain in health and disease.**

**Keywords** Colonization; Development; Microglia; Proliferation; Cytokine
**Subject Categories** Development; Neuroscience

## Introduction

Microglia are specialized immune cells that reside in the brain, constituting the first line of defense against pathogens while also fulfilling a plethora of other roles in brain development and function during both health and disease (Bridlance and Thion, 2023; Colonna and Butovsky, 2017; Cserep et al, 2021; Hammond et al, 2018; Hoeffel and Ginhoux, 2018; Li and Barres, 2018; Mosser et al, 2017; Prinz et al, 2021; Sierra et al, 2024; Thion and Garel, 2020; Thion et al, 2018a). Microglia constantly interact with the neural tissue, monitoring and responding to neuronal activity, and regulating neural networks (Badimon et al, 2020; Merlini et al, 2021; Parkhurst et al, 2013; Schafer et al, 2012; Tremblay et al, 2010; Umpierre and Wu, 2021; Wake et al, 2009; York et al, 2018). During development, not only microglia have been implicated in the regulation of synaptic function and maturation (Matera et al, 2025; Miyamoto et al, 2016; O'Keeffe et al, 2025; Paolicelli et al, 2011; Schafer et al, 2012; Weinhard et al, 2018), but also they play essential roles in cell death and survival (Cunningham et al, 2013; Hagemeyer et al, 2017; Nemes-Baran et al, 2020; Shigemoto-Mogami et al, 2014), axon tract formation (Hagemeyer et al, 2017; Pont-Lezica et al, 2014; Squarzoni et al, 2014; Wlodarczyk et al, 2017), wiring of inhibitory circuits (Favuzzi et al, 2021; Gallo et al, 2022; Gesuita et al, 2022; Squarzoni et al, 2014; Thion et al, 2019), and the maintenance of tissue integrity during brain morphogenesis (Lawrence et al, 2024). Consistently, microglial dysfunctions are associated with almost all brain pathologies, from neurodevelopmental to neurodegenerative diseases (Li and Barres, 2018; Mosser et al, 2017; Prinz and Priller, 2017; Thion and Garel, 2020; Thion et al, 2018a). Thus, understanding the roles of microglia and the key regulators of microglial functions across time and space is of the utmost importance.

[1]Centre Interdisciplinaire de Recherche en Biologie (CIRB), Collège de France, Université PSL, CNRS, INSERM, 75005 Paris, France. [2]Institut de Biologie de l'École Normale Supérieure (IBENS), École Normale Supérieure, CNRS, INSERM, Université PSL, 75005 Paris, France. [3]Sorbonne Université, Collège Doctoral, F-75005 Paris, France. [4]Department of Biochemistry and Structural Biology, University of Texas Health Science Center, San Antonio, TX 78229-3900, USA. [5]Instituto de Neurociencias de Alicante, Universidad Miguel Hernández-Consejo Superior de Investigaciones Científicas (UMH-CSIC), San Juan de Alicante, Alicante, Spain. [6]Institute of Experimental Immunology, University of Zurich, Zurich, Switzerland. [7]INSERM U1015, Gustave Roussy Cancer Campus, Villejuif 94800, France. [8]Singapore Immunology Network (SIgN), Agency for Science, Technology and Research, Singapore 138648, Singapore. [9]VIB-KU Leuven Center for Brain & Disease Research, 3000 Leuven, Belgium. [10]KU Leuven, Department of Neurosciences, Leuven Brain Institute, 3000 Leuven, Belgium. [11]Collège de France, Université PSL, 75005 Paris, France. [12]These authors contributed equally: Sonia Garel, Morgane Sonia Thion.
✉E-mail: sonia.garel@bio.ens.psl.eu; morgane.thion@college-de-france.fr

Studies over the last decades have shed light on the ontogeny of microglia: arising from myeloid progenitors in the yolk sac, microglial progenitors enter in the brain as early as embryonic day (E)9 in mice, or gestational week 4 in humans (Ginhoux et al, 2010; Menassa and Gomez-Nicola, 2018; Monier et al, 2007; Verney et al, 2010). Yolk sac-derived macrophages also generate border-associated macrophages (BAMs), which largely remain in the meninges and choroid plexus during prenatal life before spreading to the perivascular spaces postnatally (Goldmann et al, 2016; Utz et al, 2020). During embryonic development, the microglial population progressively expands through redistribution and proliferation (Alliot et al, 1999; Barry-Carroll et al, 2023; Dalmau et al, 2003; Li et al, 2019; Matcovitch-Natan et al, 2016; Menassa and Gomez-Nicola, 2018; Nikodemova et al, 2015; Ostrem et al, 2024), gradually colonizing the brain via a highly stereotyped spatiotemporal pattern with conserved features amongst mammalian species (Menassa et al, 2022; Squarzoni et al, 2014; Swinnen et al, 2013). Notable features of this process include the transient exclusion of microglia from the developing cortical plate and their accumulation at several "hotspots" (Hattori et al, 2020; Nemes-Baran et al, 2020; Squarzoni et al, 2014; Swinnen et al, 2013). Microglia progressively mature in the brain, in close interaction with their neural environment (Bennett et al, 2018; Matcovitch-Natan et al, 2016; Stogsdill et al, 2022; Thion et al, 2018b) and, in the adult brain, they homogeneously tile the brain parenchyma and maintain appropriate densities across regions (Askew et al, 2017; De Biase et al, 2017; Tay et al, 2017).

Alongside this developmental colonization, microglia exhibit considerable morphological and cellular diversity and exist in different states (Bian et al, 2020; Hammond et al, 2019; Hope et al, 2020; Kracht et al, 2020; La Manno et al, 2021; Lawrence et al, 2024; Li et al, 2019; Masuda et al, 2019; Paolicelli et al, 2022; Silvin et al, 2022; Stogsdill et al, 2022; Stratoulias et al, 2023) which are also observed during aging and in neurodegenerative conditions (Bisht et al, 2016; Keren-Shaul et al, 2017; Krasemann et al, 2017; Marschallinger et al, 2020; Masuda et al, 2019; Safaiyan et al, 2016; Sala Frigerio et al, 2019; Silvin et al, 2022). In the developing brain, this heterogeneity has begun to be associated with their distribution patterns and functions. For example, ameboid microglia transiently found in the early postnatal white matter and originally called "fountains" of microglia (Imamoto and Leblond, 1978; Verney et al, 2010) have been termed axon tract-associated microglia (ATM), proliferative-region-associated microglia (PAM) or youth-associated microglia (YAM) and participate in the regulation of gliogenesis and myelination (Hagemeyer et al, 2017; Hammond et al, 2019; Li et al, 2019; Nemes-Baran et al, 2020; Silvin et al, 2022; Wlodarczyk et al, 2017). ATM-like cells are also present during prenatal life in transient hotspots of microglial accumulation, where these microglia maintain tissue integrity in the context of morphogenetic constraints (Lawrence et al, 2024). Yet, how early microglial distribution and heterogeneity are regulated in close dialogue with the neural tissue remains largely to be investigated.

Attempts to understand the regulation of brain macrophage development, colonization and maturation have identified a key role for the colony-stimulating factor 1 receptor (CSF1R) pathway. Microglia and BAMs both express the CSF1R, which is required for their development, differentiation and survival, while its ligands, CSF-1 and interleukin-34 (IL34) are predominantly expressed by neural cells in mice (Greter et al, 2012; Nandi et al, 2012; Stanley and Chitu, 2014; Wei et al, 2010). In the adult brain, CSF-1 is mainly secreted by astrocytes and oligodendrocytes, and is required for white matter and cerebellar microglia, while IL34 produced by neurons is important for gray matter microglia (Badimon et al, 2020; Easley-Neal et al, 2019; Kana et al, 2019; Nandi et al, 2012; Wei et al, 2010). However, while microglia largely rely on CSF-1 (Cecchini et al, 1994; Chitu and Stanley, 2017; Easley-Neal et al, 2019; Ginhoux et al, 2010; Greter et al, 2012; Nandi et al, 2012; Wei et al, 2010) during embryogenesis, IL34 plays key roles postnatally (Devlin et al, 2025; Greter et al, 2012; Wang et al, 2012). ATM can also express their own *Csf1* (Benmamar-Badel et al, 2020; Hagemeyer et al, 2017; Hammond et al, 2019; Hristova et al, 2010; Li et al, 2019; Wlodarczyk et al, 2017), though the role of this expression has not yet been investigated. Despite these insights, several aspects remain incompletely characterized, such as when and where microglia proliferate during forebrain colonization. More specifically, the influence of early neuronal activity, as well as the respective contributions of distinct cellular sources of CSF-1 to the spatial and temporal control of microglial positioning and expansion, have not been clearly defined.

Here, we combined immunofluorescence analysis, live imaging, transcriptomics and genetically modified mouse models to generate a detailed timeline of microglial distribution and proliferation patterns in the forebrain, with a particular focus on the neocortex. Our findings show that microglial colonization is driven by temporally distinct waves of local proliferation, largely independent from early neuronal activity, and highlight specific hotspots where microglia accumulate in an ATM state. We also revealed that microglia regulation involves both extrinsic and intrinsic *Csf1* contributions. Neural progenitors, and to a lesser extent post-mitotic neurons, provide extrinsic *Csf1* with transient, local and dose-dependent actions, while ATM sustains their proliferation by producing intrinsic *Csf1*, a hallmark of this state. Our study sheds new light on how microglia colonize the developing forebrain by relying on distinct, local and cell-type-specific *Csf1* sources to adopt their complex pattern of distribution and proliferation, highlighting a remarkable crosstalk between *Csf1* sources, microglial niches and microglial states. Together, these findings thus add a previously unrecognized layer of spatial and temporal complexity to the regulation of microglial development by CSF1R signaling.

## Results

### Dynamic waves and local hotspots of proliferation shape microglial colonization

To investigate how microglia proliferate and are distributed during forebrain development, we used *Cx3cr1^{gfp/+}* mice (Jung et al, 2000) to label macrophages in brains from embryos (E12.5, E14.5, E16.5, and E18.5), pups (postnatal day (P)0, P3, P5, P7, P9, and P14), juveniles (P20) and adults (P60). BAMs were excluded from the analysis based on their location, morphology and lower GFP intensity (Appendix Fig. S1A,B). Using immunofluorescence for GFP, the proliferation marker Ki67 and injections of 5-ethynyl-2'-deoxyuridine (EdU) to label cells undergoing DNA replication (Appendix Fig. S1C,D), we quantified microglial density and proliferation in the somatosensory neocortex, striatum, hippocampus and preoptic area (POA) (Fig. 1A,B). We observed two broad

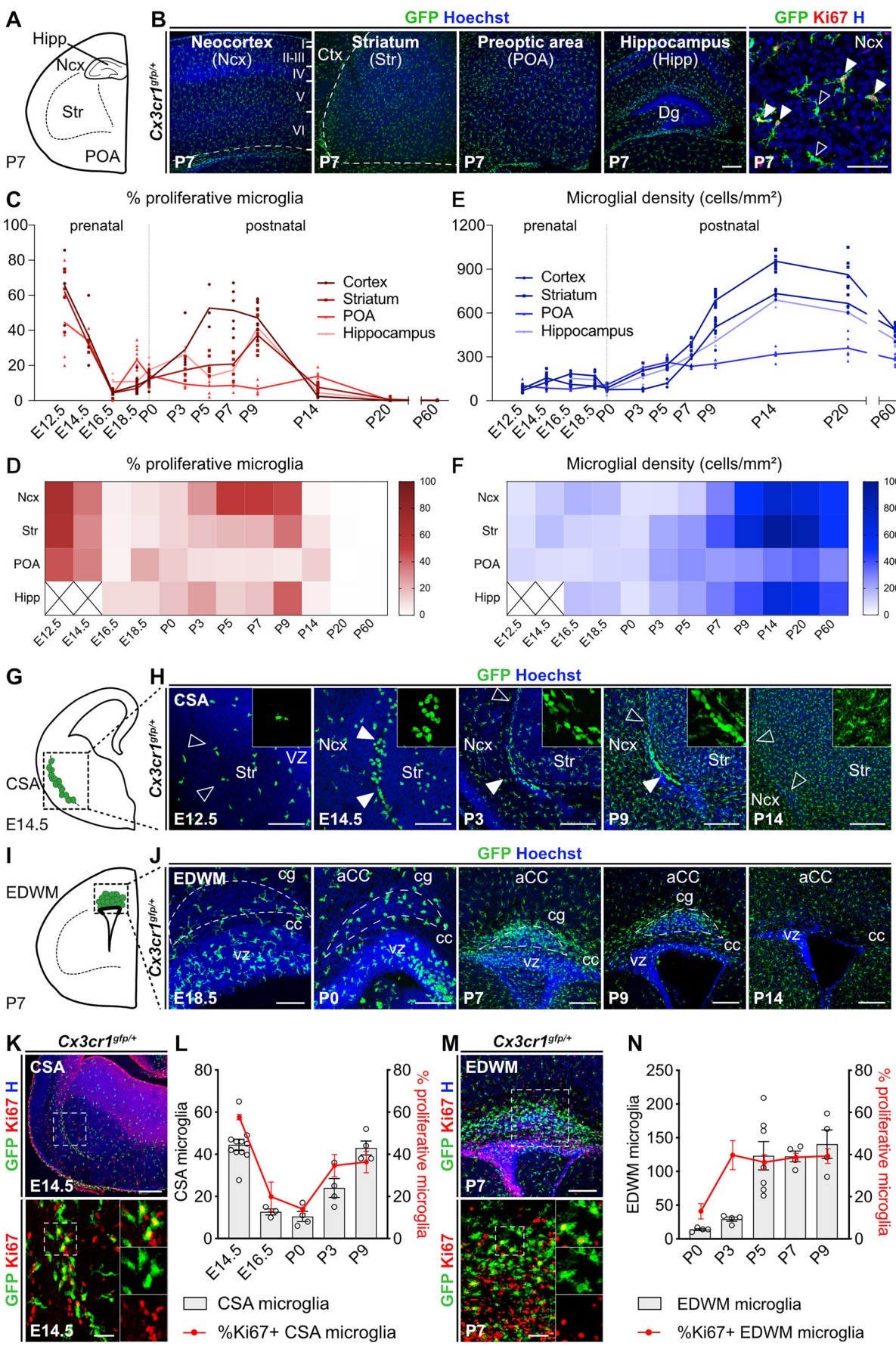

◀  **Figure 1.  Microglial brain colonization is shaped by two dynamic waves of proliferation and the emergence of developmental hotspots.**

(A) Schematic representation of a hemicoronal section of a P7 mouse telencephalon with the four regions of interest labeled. (B) Immunolabeling of coronal brain sections from P7 *Cx3cr1$^{gfp/+}$* pups showing microglia (GFP) in the somatosensory cortex, striatum, preoptic area and hippocampus (CA1 and CA3). High magnification shows co-expression of Ki67 and GFP (closed arrowheads) or absence of Ki67 expression in GFP-positive cells (open arrowheads). For the postnatal somatosensory cortex, quantification was performed on all the cortical layers (I to VI). High magnification images track single microglia dividing over the course of imaging (open arrowheads). (C, D) Percentage of proliferative (Ki67-positive) microglia among total microglia, and microglial density (cells/mm$^2$) in each region of interest from E12.5 to P60 ($n_{E12.5} = 3$ to 5; $n_{E14.5} = 3$ to 9; $n_{E16.5} = 3$ to 8; $n_{E18.5} = 3$ to 10; $n_{P0} = 5$ to 8; $n_{P3} = 4$; $n_{P5} = 3$ to 7; $n_{P7} = 4$ to 6; $n_{P9} = 3$ to 11; $n_{P14} = 4$ to 9; $n_{P20} = 3$ to 8; $n_{P60} = 8$). The curves plotted for each region are nonparametric regressions. (E, F) Heatmaps of the proportion of proliferative (Ki67-positive) microglia and microglial density (cells/mm$^2$) in each region from E12.5 to P60 ($n_{E12.5} = 3$ to 5; $n_{E14.5} = 3$ to 9; $n_{E16.5} = 3$ to 8; $n_{E18.5} = 3$ to 10; $n_{P0} = 5$ to 8; $n_{P3} = 4$; $n_{P5} = 3$ to 7; $n_{P7} = 4$ to 6; $n_{P9} = 3$ to 11; $n_{P14} = 4$ to 9; $n_{P20} = 3$ to 8; $n_{P60} = 8$). (G) Schematic representation of a hemicoronal section of an E14.5 mouse telencephalon with the CSA accumulation. (H) Immunolabeling of coronal brain sections from *Cx3cr1$^{gfp/+}$* mice showing ameboid microglia accumulating at the CSA during development. Open arrowheads show areas lacking microglia and solid arrowheads show their accumulation. The inset shows microglial close-ups with GFP only, and are 150 µm wide. (I) Schematic representation of a hemicoronal section of a P7 mouse telencephalon with the EDWM accumulation. (J) Immunolabeling of coronal brain sections from *Cx3cr1$^{gfp/+}$* mice showing accumulating ameboid microglia at the EDWM during development. Dotted lines delineate the zone of cellular accumulation (intense Hoechst staining) between the cingulum bundle and the ventral part of the corpus callosum, where quantifications were made. (K) Immunolabeling of coronal brain sections from E14.5 *Cx3cr1$^{gfp/+}$* embryos showing co-expression of GFP and Ki67 proliferative cells at the CSA. Close-ups are 75 µm wide. (L) Number of ameboid microglia and proportion of proliferative (Ki67-positive) ameboid microglia at the CSA from E14.5 to P9 ($n_{E14.5} = 11$ for microglia number and $n_{E14.5} = 6$ for microglial proliferation; $n_{E16.5} = 3$; $n_{P0} = 4$; $n_{P3} = 4$; $n_{P9} = 4$). (M) Immunolabeling of coronal brain sections from P7 *Cx3cr1$^{gfp/+}$* pups showing co-expression of GFP and Ki67 proliferative cells at the EDWM. Close-ups are 75 µm wide. (N) Number of ameboid microglia and proportion of proliferative (Ki67-positive) ameboid microglia at the EDWM from P0 to P9 ($n_{P0} = 4$; $n_{P3} = 4$; $n_{P5} = 7$; $n_{P7} = 4$; $n_{P9} = 4$). Data were presented as mean ± SEM (L, M). Scale bars: 250 µm in (B, low mag), (H, P3 to P14) and (K, low mag); 100 µm in (B, high mag) and 100 µm (J, E18.5); 150 µm in (H, E12.5-E14.5) and (J, E18.5); 200 µm in (J, P3-P14) and (M, low mag); 50 µm in (K, high mag) and (M, high mag). aCC anterior cingulate cortex, cc corpus callosum, cg cingulum bundle, CSA cortico-striato-amygdalar boundary, Dg dentate gyrus, EDWM early-dorsal white matter, Hipp hippocampus; Ncx neocortex, POA preoptic area, Str striatum, VZ ventricular zone. See also Appendix Fig. S1 and EV1. Source data are available online for this figure.

waves of proliferation associated with increased density in both males and females (Fig. 1C–F). During the first proliferative wave, at E12.5, around 60% of microglia were actively proliferating (Fig. 1C–F), about half of these cells displayed sustained proliferation until E14.5 (Fig. 1C,D), after which proliferation drastically decreased, reaching an average of 5% by E16.5 (Fig. 1C,D). Despite the high level of proliferation, microglial density remained stable during this first wave (Fig. 1E,F), likely due to concurrent brain growth. The second wave occurred between P3 and P9, with variations in timing and intensity according to the region (Fig. 1C,D). In the neocortex, striatum and hippocampus, density peaked at P14, followed by a decrease towards adult level (Fig. 1E,F), whereas the POA postnatal proliferation remained at low levels, resulting in stable density over time (Fig. 1C–F).

We also examined developmental "hotspots", where microglia transiently accumulate (Hristova et al, 2010; Imamoto and Leblond, 1978; Kershman, 1939; Lawrence et al, 2024; Verney et al, 2010), focusing on the embryonic cortico-striato-amygdalar boundary (CSA) (Fig. 1G,H) and the postnatal early-dorsal white matter (EDWM) (Fig. 1I,J), which lies deep to the anterior cingulate cortex (Hagemeyer et al, 2017; Hammond et al, 2019; Lawrence et al, 2024; Li et al, 2019; Squarzoni et al, 2014; Wlodarczyk et al, 2017). In contrast to a mostly scattered distribution at E12.5, ameboid microglia transiently accumulated at the CSA around E14.5 (Fig. 1H). By E16.5, most microglia were ramified in this region, though a few ameboid cells accumulated ventrolaterally up to the end of the first postnatal week, and the hotspot had disappeared by P14 (Fig. 1H). In parallel, ameboid microglia started to accumulate in the neonate EDWM, peaked by the end of the first postnatal week and, again, the hotspot had disappeared by P14 (Fig. 1J), as in the CSA. In both hotspots, microglia displayed sustained proliferation, which correlated with the number of ameboid microglia (Fig. 1K–N). To assess whether hotspots may be supplying the surrounding parenchyma with microglia, we focused on the CSA and examined microglial dynamics using two-photon live-imaging on E14.5 brain slices (Fig. EV1A). Live-imaging confirmed that

microglia were actively dividing in both the neocortex and CSA (Fig. EV1B,C). While cortical microglia were very mobile as previously shown (Smolders et al, 2017; Swinnen et al, 2013), CSA microglia had a reduced mean speed and remained local, suggesting that they do not act as a reservoir for surrounding regions (Fig. EV1B,D–F).

Taken together, microglial expansion follows two waves of proliferation: a first widespread intense phase ending around mid-neurogenesis, and a second postnatal one whose intensity and timing vary across regions. Microglial colonization is further shaped by transient hotspots of proliferating microglia that display a distinctive behavior and morphology.

## Early neuronal activity does not disrupt the dynamics of cortical microglial colonization

To deepen our understanding of the underlying mechanisms regulating microglial proliferation and distribution, we focused on the somatosensory cortex in which the pattern of microglial colonization is highly stereotypical and heterogeneous (Arnoux et al, 2013; Hattori et al, 2020; Squarzoni et al, 2014) (Fig. 2A). Before E16.5, microglial proliferation was homogenous with cells accumulating in the subplate and largely excluded from the cortical plate (Fig. 2A–F). As observed using live imaging, only a few microglial cells crossed the cortical plate towards the marginal zone at E14.5 (Fig. EV1B). Perinatally, microglia progressively colonized deep layers and then, postnatally, the upper layers, with a consistent shift of proliferation from deeper to upper which correlates with a homogenization of microglial densities across layers (Fig. 2A–C). By P14, microglia were homogeneously distributed in the cortical plate, and proliferation had dropped (Fig. 2A–F).

Since microglia sense neuronal activity (Duran Laforet & Schafer, 2024; Nimmerjahn et al, 2005; Schafer et al, 2012; Umpierre and Wu, 2021) which changes along developmental phases (Cossart and Garel, 2022; Molnar et al, 2020), and activity has been recently shown to regulate microglial neocortical density

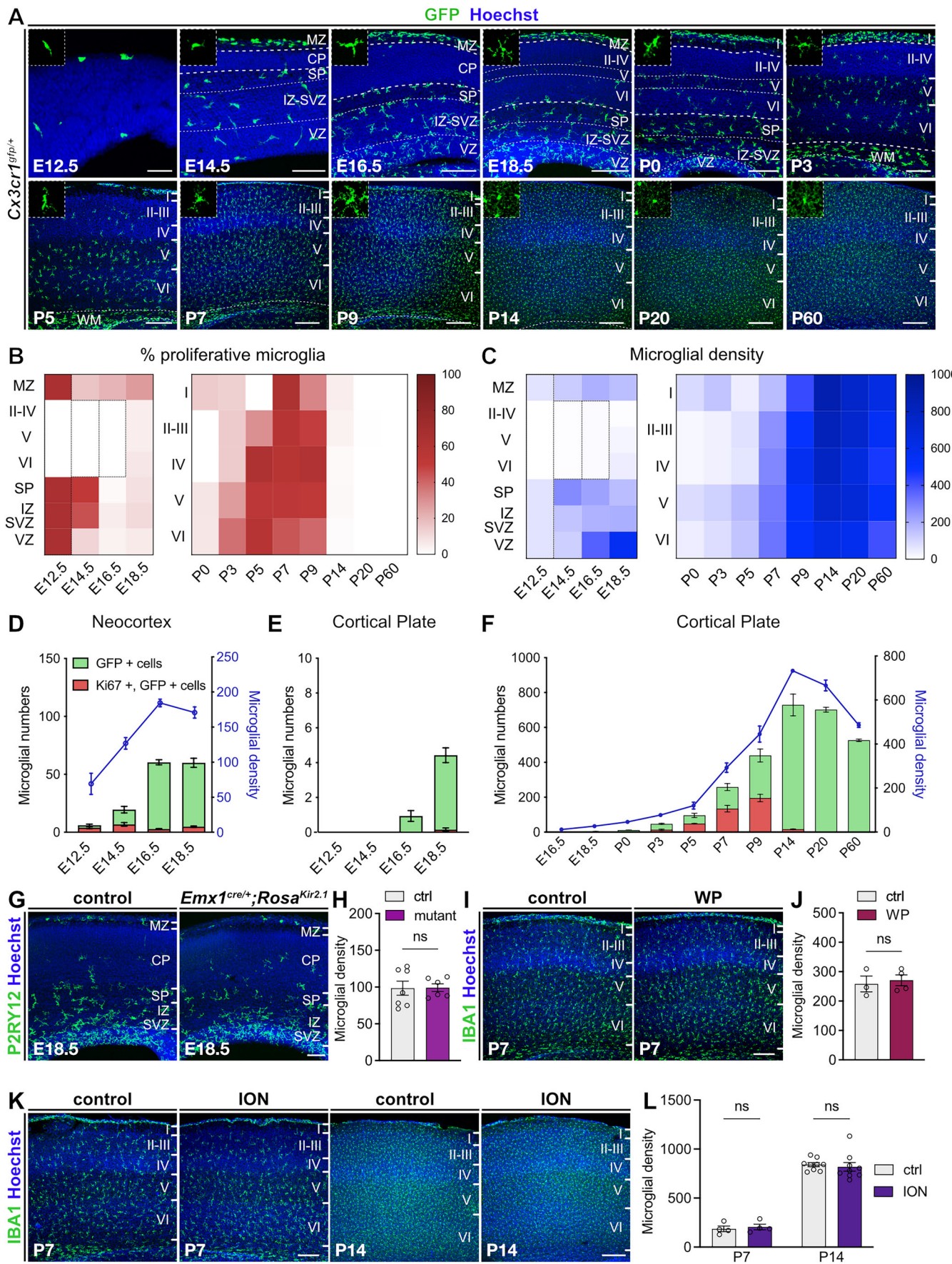

**Figure 2.  Timely microglial colonization of the cerebral cortex remains unaffected by perturbations of neuronal activity.**

(A) Immunolabeling of coronal brain sections from $Cx3cr1^{gfp/+}$ mice showing microglial (GFP) colonization in the somatosensory cortex from E12.5 to P60. Until P3, the cortical plate is underlined by bold dotted lines. Microglial close-ups, delineated by dotted lines, are 75 µm wide. The image at P7 is the same used in Fig. 1B. (B, C) Heatmaps of the proportion of proliferative (Ki67-positive) microglia and of microglial density (cells/mm²) in each cortical layer from E12.5 to P60 ($n_{E12.5} = 4$, $n_{E14.5} = 9$, $n_{E16.5} = 8$, $n_{E18.5} = 10$, $n_{P0} = 8$, $n_{P3} = 4$, $n_{P5} = 3$, $n_{P7} = 6$, $n_{P9} = 11$, $n_{P14} = 9$, $n_{P20} = 8$, $n_{P60} = 8$). In each case, the heatmap on the left represents all layers of the embryonic neocortex, while the heatmap on the right shows postnatal cortical layers (I to VI). Dotted rectangles correspond to layers not distinguished for quantification. At E12.5, the quantifications were made without distinguishing between different layers. (D) Number of Ki67-negative and Ki67-positive microglia as well as microglial density (blue line, cells/mm²) in all layers of the neocortex of $Cx3cr1^{gfp/+}$ embryos ($n_{E12.5} = 4$, $n_{E14.5} = 9$, $n_{E16.5} = 8$, $n_{E18.5} = 10$). (E) Microglial numbers in the cortical plate (layers II to VI) of the neocortex of $Cx3cr1^{gfp/+}$ embryos ($n_{E12.5} = 4$, $n_{E14.5} = 9$, $n_{E16.5} = 8$, $n_{E18.5} = 10$). (F) Number of Ki67-negative and Ki67-positive microglia as well as microglial density (blue line, cells/mm²) across cortical layers (layers II to VI for prenatal stages, layers I to VI for postnatal stages) of $Cx3cr1^{gfp/+}$ mice from E16.5 to P60 ($n_{E16.5} = 8$, $n_{E18.5} = 10$, $n_{P0} = 8$, $n_{P3} = 4$, $n_{P5} = 3$, $n_{P7} = 6$, $n_{P9} = 11$, $n_{P14} = 9$, $n_{P20} = 8$, $n_{P60} = 8$). (G) Immunolabeling of coronal brain sections from E18.5 control and $Emx1^{cre/+};Kir2.1$ showing microglia (P2RY12-positive) in the somatosensory cortex. (H) Microglial density (IBA1-positive cells/mm²) in associated control and $Emx1^{cre/+};Kir2.1$ (mutant) at E18.5 ($n_{control} = 7$; $n_{mutant} = 6$; from two distinct litters). (I) Immunolabeling of coronal brain sections from P7 control mice and mice having undergone bilateral daily whisker plucking from P1 to P7 (WP), showing microglia (IBA1-positive) in the somatosensory cortex. (J) Microglial density (IBA1-positive cells/mm²) in associated control and WP mice at P7 ($n_{control} = 4$; $n_{WP} = 3$; from one litter). (K) Immunolabeling of coronal brain sections from P7 and P14 control mice and mice having undergone unilateral infraorbital nerve section performed at P1 (controls are the cortex of the ipsilateral hemisphere and ION of the contralateral hemisphere). (L) Microglial density (IBA1-positive cells/mm²) in control and ION mice at P7 (littermate control, $n = 4$ from two distinct litters) and P14 (littermate control, $n = 9$ from three distinct litters). Data were presented as mean ± SEM. Two-sided unpaired Mann–Whitney tests (H, I) or two-sided Wilcoxon matched-pairs signed-rank tests for each stage (L) were performed to assess differences in microglial density. ns not significant. Scale bars: 50 µm (A, E12.5); 100 µm (A, E14.5); 200 µm (A, E16.5, E18.5, P0 and P3); 250 µm (A, P5 to P60) and (K, P14); 200 µm (G, I) and (K, P7). CP cortical plate, ctrl control, ION infraorbital nerve section, IZ intermediate zone, MZ marginal zone, SP subplate, SVZ subventricular zone, VZ ventricular zone, WM white matter, WP whisker pluck; I, II, III, IV, V, and VI are respectively cortical layers I, II, III, IV, V, and VI. See also Fig. EV1; Appendix Fig. S2. Source data are available online for this figure.

in the second postnatal week (Kumaraguru et al, 2025), we wondered whether neuronal activity could contribute to early shaping of microglial numbers and distribution. Focusing on the well-studied somatosensory cortex, we first targeted prenatal activity by crossing the Cre-dependent $Rosa^{Kir2.1}$ line (Moreno-Juan et al, 2017) with the $Emx1^{cre/+}$ mouse line, thereby driving hyperpolarization of the whole cortical primordium (Anton-Bolanos et al, 2019). Since we did not find any alterations in microglial density or numbers in E18.5 $Emx1^{cre/+};Kir2.1$ mice compared to controls (Fig. 2G,H; Appendix Fig. S2A), we next tested evoked sensory activity, which arises during the first postnatal week, concurrent with intense microglial proliferation and an increase in their density (Fig. 2A–F). To this aim, we performed either daily bilateral whisker plucking (WP) or unilateral infraorbital nerve section (ION) at P1. These procedures modulate sensory inputs coming from the whisker pad to the somatosensory cortex in distinct ways, with only ION leading to marked morphological changes in somatosensory cortex barrels (White et al, 1990). Neither of these perturbations significantly affected microglial density or population numbers at P7 (Fig. 2I–L; Appendix Fig. S2B,C), nor was their distribution at P7 or P14 altered (Appendix Fig. S2D–G).

Altogether, these analyses provide insights into microglial distribution and proliferation within the neocortex and suggest that, in contrast to reports at later stages (Kumaraguru et al, 2025), surrounding neuronal activity is not a major driver of microglial colonization during early development.

## Local sources of CSF-1 are essential for cortical embryonic microglia

Since neuronal activity was not a main driver of early microglial colonization, we focused on CSF1R signaling, examining the sources of its ligands—CSF-1 and IL34—within the early neural environment in both mouse and human (Fig. EV2A–F). We first explored a publicly available longitudinal dataset of single-cell RNA sequencing (scRNA-seq) of the mouse neocortex (E10-P4)(Di Bella

et al, 2021) (Fig. EV2A,B), and found that $Csf1$ expression by neural progenitors increases during development, with migrating and immature neurons also beginning to express $Csf1$ from E14 onwards (Fig. EV2A). At later stages, $Csf1$ starts to be expressed by oligodendrocytes and more prominently by astrocytes (Fig. EV2A), as reported in adult mice (Badimon et al, 2020; Kana et al, 2019).

To assess whether these expression patterns are conserved in humans, we analyzed two complementary datasets: first, HuMous.org, an interactive platform providing longitudinal scRNA-seq data from radial glia and neurons in the prenatal mouse and human neocortex (Javed et al, 2025) (Fig. EV2C); and second, a postnatal single-nucleus RNA sequencing dataset from human prefrontal and to a lesser extent, parietal cortex, covering individuals aged 4 to 19 years (Baldassari et al, 2025; Velmeshev et al, 2023) (Fig. EV2D–F). Consistent with our observations in mice, the highest $Csf1$ expression in humans occurs in prenatal progenitors (Fig. EV2C). However, in the postnatal human cortex, $Csf1$ expression is no longer detected in neurons, astrocytes, or oligodendrocytes, suggesting a developmentally restricted expression pattern in humans (Fig. EV2D,E). In mice, $Il34$ instead was mainly found from several days after birth in mature neurons of deep cortical layers, consistent with its known postnatal expression (Di Bella et al, 2021; Nandi et al, 2012) (Fig. EV2B). In humans, while its expression is not detected prenatally in the HuMous dataset, it is clearly expressed postnatally in excitatory neurons, consistent with a role emerging after birth. Importantly, $Csf1r$ transcripts are not detected in mouse or human neural progenitors or post-mitotic neurons during prenatal life (Fig. EV2D–F). We also confirmed using RNAscope that $Csf1r$ transcripts are not observed in non-microglial cells in the E14.5 mouse neocortex (Fig. EV2G).

To investigate the respective contributions of each CSF1R ligand, we used genetic tools to conditionally inactivate $Csf1$ or $Il34$ in distinct cell populations. First, we targeted $Csf1$ expression in neocortical progenitors and their progeny, including excitatory neurons, and later astrocytes and oligodendrocytes, using $Emx1^{cre}$; $Csf1^{fl/fl}$ mice (Gorski et al, 2002) (Fig. 3A,B). This approach led to a

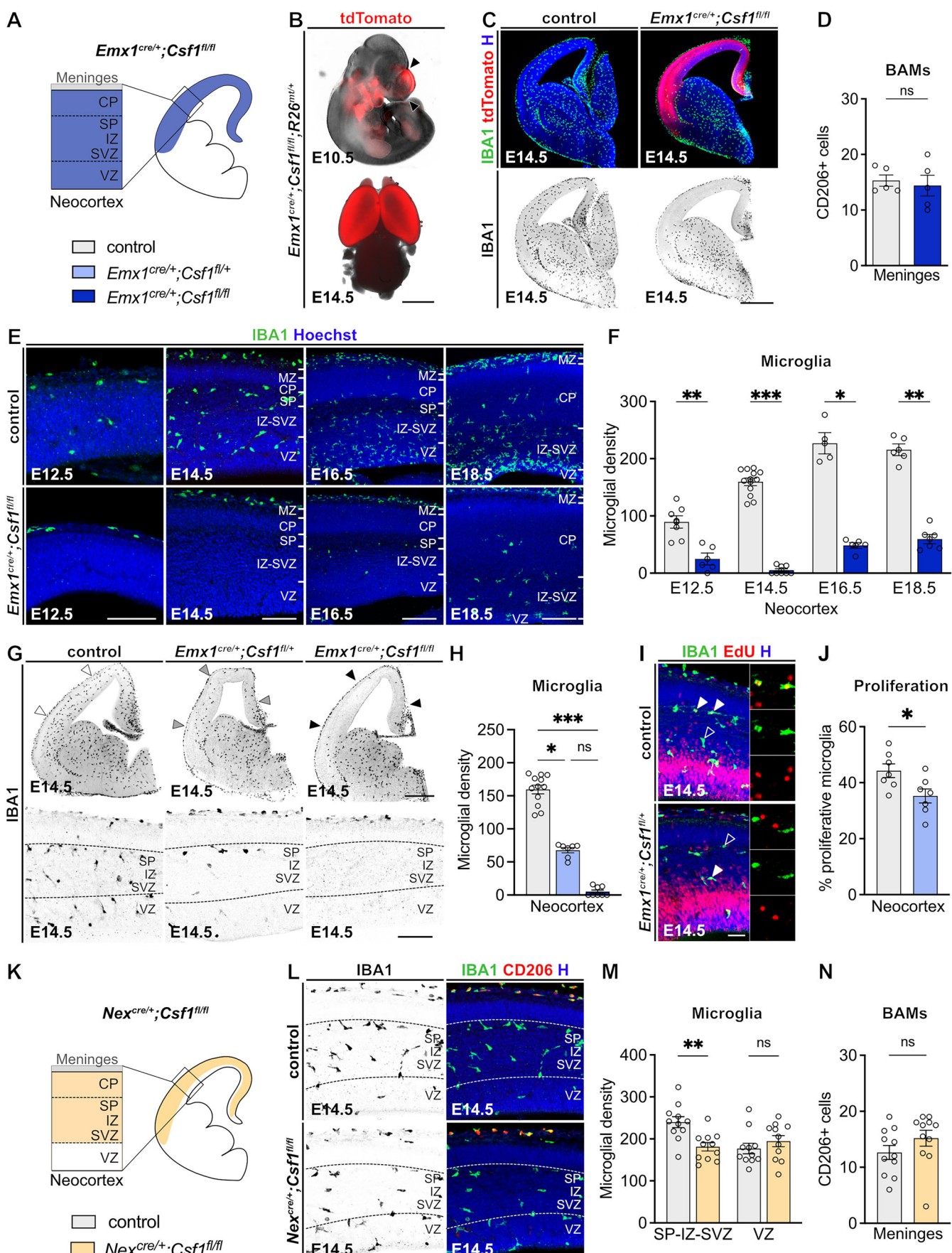

**Figure 3. Cortical CSF-1 is required for microglial colonization and proliferation in a local, specific and dose-dependent manner.**

(A) Schematic representation of a hemicoronal section of an E14.5 mouse telencephalon showing the pattern of recombination driven by the $Emx1^{cre}$ line in the embryonic brain, including all progenitors and excitatory neurons of the neocortex. (B) Immunolabeling of a whole embryo at E10.5 or of the whole brain at E14.5 showing the recombination pattern driven by the $Emx1^{cre}$ line (tdTomato) in $Emx1^{cre/+}$;$Csf1^{fl/fl}$; $R26^{mt/+}$ embryos. (C) Immunolabeling of coronal brain sections from E14.5 control and $Emx1^{cre/+}$;$Csf1^{fl/fl}$; $R26^{mt/+}$ mice showing IBA1 and tdTomato expression and highlighting the depletion of microglia in conditional mutants restricted to the area of the $Emx1^{cre}$ pattern of expression (tdTomato). (D) Number of BAMs in the meninges (over a 500 µm length) of the neocortex at E14.5 ($n_{control} = 5$, $n_{homo} = 5$; from two distinct litters). (E) Immunolabeling of coronal brain sections from E12.5 to E18.5 in control and $Emx1^{cre/+}$;$Csf1^{fl/fl}$ embryos showing IBA1-positive cells in the neocortex. (F) Microglial density (IBA1-positive cells/mm$^2$) in the neocortex of control and $Emx1^{cre/+}$;$Csf1^{fl/+}$ embryos (E12.5: $n_{control} = 7$, $n_{homo} = 5$; E14.5: $n_{control} = 12$, $n_{homo} = 8$; E16.5: $n_{control} = 4$, $n_{homo} = 5$; E18.5: $n_{control} = 5$, $n_{homo} = 6$; from at least two distinct litters per stage)(control vs homo E12.5: $P = 0.0051$; E14.5: $P < 0.0001$; E16.5: $P = 0.0159$; E18.5: $P = 0.0043$). (G) Immunolabeling of coronal brain sections from E14.5 control, $Emx1^{cre/+}$;$Csf1^{fl/+}$ and $Emx1^{cre/+}$;$Csf1^{fl/fl}$ embryos showing IBA1 expression and highlighting the local depletion of microglia in heterozygotes (gray arrowheads) and homozygotes (black arrowheads). IBA1 also stains BAMs in the meninges. (H) Microglial density (IBA1+ cells/mm$^2$) in the neocortex at E14.5 ($n_{control} = 12$; $n_{hetero} = 7$; $n_{homo} = 8$; from at least three distinct litters)(control vs hetero: $P = 0.0348$; control vs homo: $P < 0.0001$; hetero vs homo $P = 0.2011$). (I) Immunolabeling of coronal brain sections from E14.5 control and $Emx1^{cre/+}$;$Csf1^{fl/fl}$ embryos showing expression of IBA1 among EdU-positive cells. Open arrowheads highlight IBA1+, EdU- cells, black arrowheads highlight IBA1+, EdU+ cells. (J) Percentage of proliferative microglia in control and $Emx1^{cre/+}$;$Csf1^{fl/fl}$ E14.5 embryos ($n_{control} = 7$, $n_{homo} = 7$; from two distinct litters) ($P = 0.0379$). (K) Schematic representation of a hemicoronal section of an E14.5 mouse telencephalon showing the pattern of expression of the $Nex^{cre}$ line in the embryonic brain, including excitatory neurons of the neocortex but not progenitors. (L) Immunolabeling of coronal brain sections from E14.5 control and $Nex^{cre/+}$;$Csf1^{fl/fl}$ embryos showing IBA1 and CD206-positive cells in the neocortex. (M) Microglial densities in the neocortex at E14.5 ($n_{control} = 11$, $n_{homo} = 11$; from three distinct litters) (SP-IZ-SVZ: $P = 0.0032$; VZ: $P = 0.5302$). (N) Number of BAMs in the meninges (over a 500 µm length) of the neocortex at E14.5 ($n_{control} = 11$, $n_{homo} = 11$; from three distinct litters). Data were presented as mean ± SEM. Two-sided unpaired Mann–Whitney test (D, J, N) or Kruskal–Wallis with Dunn's post hoc test (H) or two-way ANOVA with Sidak's post hoc test (M) were performed to assess differences. ns, not significant, * $P < 0.05$, ** $P < 0.01$, *** $P < 0.001$. Scale bars: 1 mm (B); 250 µm (C); 100 µm (E, E12.5 and E14.5) (G, high mag) and (L); 200 µm (E, E16.5 and E18.5); 500 µm (G, low mag); and 50 µm (I). BAMs border-associated macrophages, CP cortical plate, H Hoechst, IZ intermediate zone, MZ marginal zone, SP subplate, SVZ subventricular zone, VZ ventricular zone; II, III, IV, V, and VI are respectively cortical layers II, III, IV, V, and VI. See also Figs. EV2 and 4. Source data are available online for this figure.

selective change in *Csf1* expression in the neocortex and a drastic depletion of microglia restricted to the neocortex at E14.5 (Fig. EV2H), without changing microglial numbers or proliferation in adjacent regions, such as the CSA or the striatum (Fig. EV3A–C). The depletion started from E12.5 and was specific to microglia since meningeal BAM numbers remained unchanged at E14.5 (Fig. 3C–F). Interestingly, $Emx1^{cre}$; $Csf1^{fl/+}$ heterozygous mice displayed an intermediate phenotype at this stage (Fig. 3G,H), highlighting a dose-dependent effect on microglia, which was not due to the presence of the $Emx1^{cre}$ allele (Fig. EV3D–F). Furthermore, we observed a similar drastic depletion of neocortical microglia at E14.5 in $Nestin^{cre}$; $Csf1^{fl/fl}$ mice, in which a broad population of neural precursors are targeted (Tronche et al, 1999) (Fig. EV3G–I). After their local depletion in $Emx1^{cre}$; $Csf1^{fl/fl}$ embryos, few microglia were present in the neocortex from E16.5 until the end of embryogenesis (Fig. 3E,F). We thus took advantage of $Emx1^{cre}$; $Csf1^{fl/+}$ heterozygous mice, in which $Csf1$ levels are reduced but microglia can nevertheless be found in the neocortex, to assess the impact of neural $Csf1$ on microglial proliferation at E14.5. We found that the number of EdU-positive microglia were decreased in $Emx1^{cre}$; $Csf1^{fl/+}$ mice compared to controls (Fig. 3I,J), highlighting a reduced proliferation. By contrast, and consistent with a dominant prenatal role of $Csf1$, inactivation of $Il34$ in the neocortex did not have any effect on either microglial density or the number of BAMs at E14.5 (Fig. EV3J–L).

Since $Csf1$ was expressed at high levels in progenitors, but also in migrating neurons, we next aimed to disentangle the contributions of these two sources. Inactivation of $Csf1$ only in post-mitotic cortical neurons using $Nex^{cre}$; $Csf1^{fl/fl}$ mice (Fig. 3K)(Goebbels et al, 2006), led to a small but significant decrease in microglial numbers in the subplate, intermediate zone and subventricular zone, but not in the ventricular zone where apical progenitors are located (Fig. 3L,M); while there was no impact on the number of BAMs (Fig. 3N).

Altogether, our data show that embryonic microglia rely on extrinsic $Csf1$ expressed mainly by progenitors and to a lesser extent by post-mitotic neurons. $Csf1$ action is extremely local, dose-dependent and contributes to microglial proliferation, establishing the importance of local $Csf1$ rather than circulating $Csf1$. In contrast, meningeal BAM survival and recruitment are likely regulated independently of signaling from the underlying brain parenchyma.

## Cortical microglia only transiently rely on neural CSF-1

The depletion of microglia from the neocortex in $Emx1^{cre}$; $Csf1^{fl/fl}$ embryos was followed by a progressive repopulation, which was completed by P7 (Fig. 4A,B). During the repopulation process, while overall microglial density was lower in $Emx1^{cre}$; $Csf1^{fl/fl}$ animals compared to controls, microglial distribution across the cortical layers remained similar in both groups, albeit that layer IV was not fully repopulated by P7 in mutants (Appendix Fig. S3A–C). This suggests that neural-derived $Csf1$ is not essential for establishing the overall laminar organization of microglia in the cortex. Using $Emx1^{cre}$; $Csf1^{fl/fl}$; $Il34^{fl/fl}$ double conditional knockout mice, we observed a drastic reduction in microglial density at P7, highlighting the critical role of $Il34$ in microglial repopulation in $Emx1^{cre}$; $Csf1^{fl/fl}$ animals (Fig. 4C–E). To further investigate how the continuous absence of cortical $Csf1$ affects the cellular and molecular properties of microglia, we performed scRNA-seq of P3 cortices from control and $Emx1^{cre}$; $Csf1^{fl/fl}$ mice (Fig. 4F; Appendix Fig. S3D–F). Differentially expressed gene (DEG) analyses between microglia from controls and mutants highlighted decreased expression of genes encoding canonical microglial markers, including *P2ry12*, *Siglech* and *Sall1*, and increased expression of *Mrc1*, *Tmem176b* and *MEV2a6c* in mutant microglia (Fig. 4G and Table EV1). We confirmed with immunolabeling that P2RY12 was undetectable in E18.5 $Emx1^{cre}$; $Csf1^{fl/fl}$ cortical microglia, while its expression was greatly reduced by P3 (Fig. 4H). Additionally, the repopulating microglia appeared less ramified, coherently with a more immature phenotype highlighted by low P2RY12 expression (Fig. 4H).

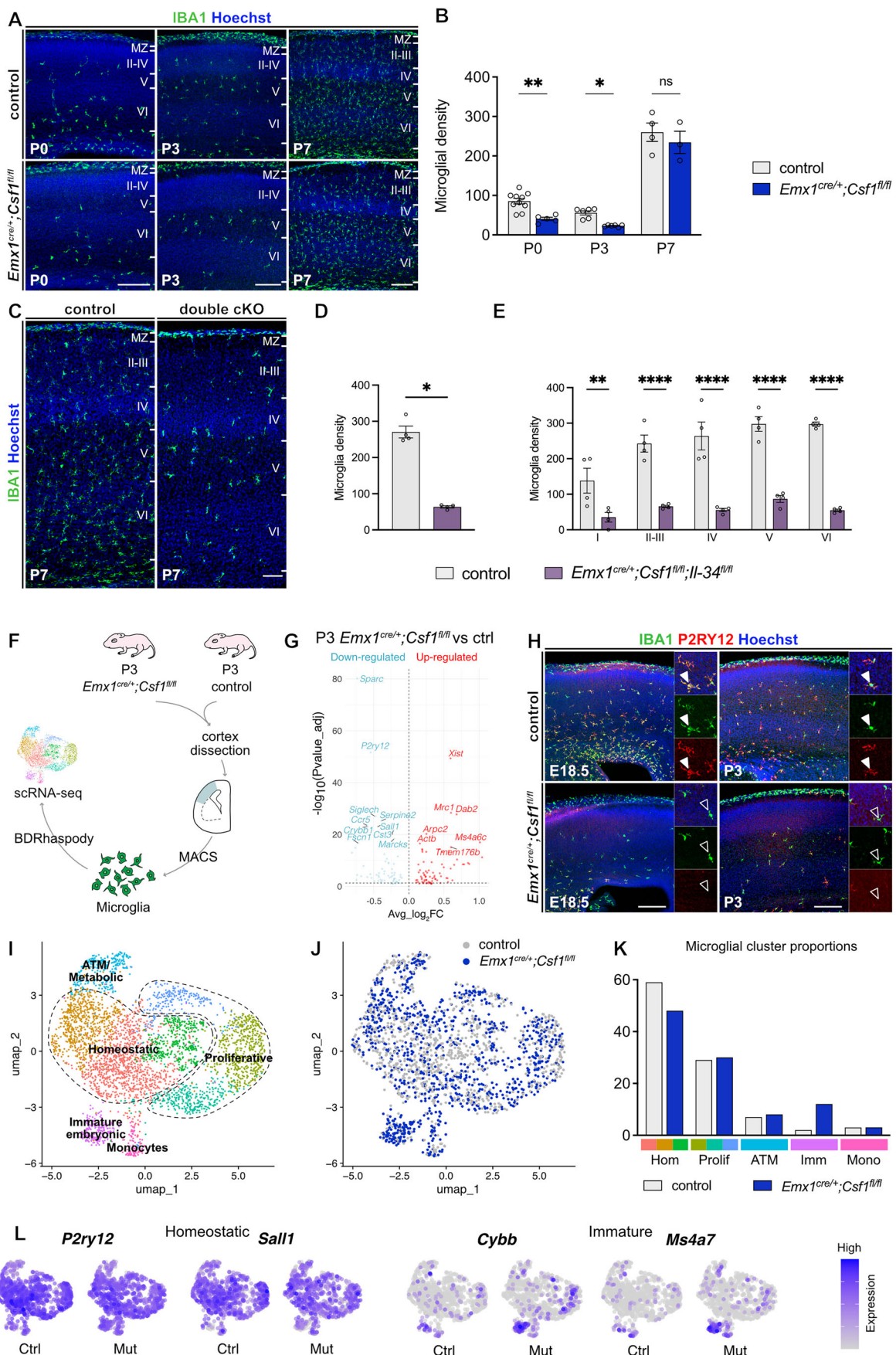

**Figure (A–E).** IBA1 Hoechst staining of control and *Emx1^{cre/+};Csf1^{fl/fl}* cortex at P0, P3, P7.

**B** Microglial density at P0, P3, P7 (control vs *Emx1^{cre/+};Csf1^{fl/fl}*). ** * ns

**C** IBA1 Hoechst, control vs double cKO, P7.

**D** Microglia density (control vs double cKO). *

**E** Microglia density per layer (I, II-III, IV, V, VI). control vs *Emx1^{cre/+};Csf1^{fl/fl};Il-34^{fl/fl}*. ** **** **** **** ****

**F** Schematic: P3 *Emx1^{cre/+};Csf1^{fl/fl}* and P3 control; cortex dissection; MACS; Microglia; BDRhaspody; scRNA-seq.

**G** P3 *Emx1^{cre/+};Csf1^{fl/fl}* vs ctrl. Down-regulated / Up-regulated volcano plot. -log₁₀(Pvalue_adj) vs Avg_log₂FC. Genes: Sparc, P2ry12, Siglech, Ccr5, Crybb1, Fscn1, Serpine2, Sall1, Cst3, Marcks, Xist, Mrc1, Dab2, Arpc2, Actb, Ms4a6c, Tmem176b.

**H** IBA1 P2RY12 Hoechst; control and *Emx1^{cre/+};Csf1^{fl/fl}* at E18.5 and P3.

**I** UMAP clusters: ATM/Metabolic, Homeostatic, Proliferative, Immature embryonic, Monocytes.

**J** UMAP control vs *Emx1^{cre/+};Csf1^{fl/fl}*.

**K** Microglial cluster proportions. Hom, Prolif, ATM, Imm, Mono. control vs *Emx1^{cre/+};Csf1^{fl/fl}*.

**L** *P2ry12*, Homeostatic *Sall1*, *Cybb*, Immature *Ms4a7*; Ctrl vs Mut. Expression High–Low.

Figure 4. Microglia recolonize the neocortex postnatally and display immature features after transient dependence on local cortical CSF-1.

(A) Immunolabeling of coronal brain sections from P0 to P7 in control and *Emx1*$^{cre/+}$;*Csf1*$^{fl/fl}$ pups showing IBA1-positive cells in the neocortex and illustrating the progressive repopulation by microglia. (B) Microglial density (IBA1-positive cells/mm²) in the neocortex of control and *Emx1*$^{cre/+}$;*Csf1*$^{fl/fl}$ pups (P0: $n_{control}$ = 10, $n_{mutant}$ = 5; P3: $n_{control}$ = 6, $n_{mutant}$ = 6; P7: $n_{control}$ = 4, $n_{mutant}$ = 3; from two distinct litters per stage) (control vs mutant P0: P = 0.0013; P3: P = 0.0022; P7: P = 0.6286). (C) Immunolabeling of coronal brain sections at P7 in control and *Emx1*$^{cre/+}$;*Csf1*$^{fl/fl}$;*Il34*$^{fl/fl}$ double cKO pups showing IBA1-positive cells in the neocortex and illustrating the drastic depletion of microglia. (D) Microglial density (IBA1-positive cells/mm²) in the neocortex of control and *Emx1*$^{cre/+}$;*Csf1*$^{fl/fl}$;*Il34*$^{fl/fl}$ P7 pups ($n_{control}$ = 4, $n_{mutant}$ = 4; from two litter) (P = 0.0286). (E) Microglial density (IBA1-positive cells/mm²) in the different layers of the neocortex of control and *Emx1*$^{cre/+}$;*Csf1*$^{fl/fl}$;*Il34*$^{fl/fl}$ P7 pups ($n_{control}$ = 4, $n_{mutant}$ = 4; from two litter) (control vs mutant: P < 0.01 in layer I; P < 0.0001 in layer II-VI). (F) Schematic representation of the workflow used to perform single-cell RNA-sequencing (scRNA-seq) on P3 control and *Emx1*$^{cre/+}$;*Csf1*$^{fl/fl}$ cortices. (G) Volcano plot of differentially expressed genes (DEGs) between P3 control and *Emx1*$^{cre/+}$;*Csf1*$^{fl/fl}$ pups in microglia (false discovery rate[FDR]-adjusted p < 0.05 and avgerage_log2FC >0.3). Genes downregulated in *Emx1*$^{cre/+}$;*Csf1*$^{fl/fl}$ pups are displayed in blue, while those upregulated are shown in red; some genes were manually annotated ($n_{control}$ = 3 (2167 cells), $n_{mutant}$ = 3 (1293 cells)). (H) Immunolabeling of coronal brain sections from E18.5 and P3 control and *Emx1*$^{cre/+}$;*Csf1*$^{fl/fl}$ pups showing co-expression of IBA1 and P2RY12 in the neocortex (closed arrowheads). Repopulating microglia were less ramified and did not express P2RY12 (open arrowheads). Close-ups are 150 μm wide. (I) UMAP visualization of scRNA-seq data representing macrophage subsets extracted from P3 control and *Emx1*$^{cre/+}$;*Csf1*$^{fl/fl}$ pups colored by annotated clusters. (J) UMAP visualization of all sorted cells colored by control (gray) or *Emx1*$^{cre/+}$;*Csf1*$^{fl/fl}$ (blue) conditions. (K) Proportions of microglial clusters among P3 control and *Emx1*$^{cre/+}$;*Csf1*$^{fl/fl}$ cortices. (L) UMAP visualization of top specific genes of homeostatic or immature clusters showing their average expression in control (Ctrl) or *Emx1*$^{cre/+}$;*Csf1*$^{fl/fl}$ (Mut) mice. Data are presented as mean ± SEM. Two-sided unpaired Mann–Whitney test were performed for each stage to assess differences (B, D); two-way ANOVA with Sidak's post hoc test was performed to assess differences (E), and Wilcoxon's rank-sum test was performed to assess differences (G). ns not significant, *P < 0.05, **P < 0.01, ***P < 0.001. Scale bars, 200 μm (A, C), 250 μm (H). ATM axon tract-associated microglia, Ctrl control, Hom homeostatic, Imm immature, IZ intermediate zone, Mono monocytes, Mut mutant, Prolif proliferative, I, II, III, IV, V, and VI are respectively cortical layers I, II, III, IV, V, and VI. See also Fig. EV2, Appendix S3, EV4 and 3, and Table EV1. Source data are available online for this figure.

To further dissect microglial molecular signatures in the cortex of *Emx1*$^{cre}$; *Csf1*$^{fl/fl}$ mice, we investigated microglial clusters and identified three clusters that expressed canonical microglial markers, including *P2ry12*, *Siglech*, *Tmem119*, and *Cx3cr1*, which we labeled "homeostatic microglia" (Figs. 4I and EV4). Homeostatic microglia were less abundant in *Emx1*$^{cre}$; *Csf1*$^{fl/fl}$ cortices (representing 47% of all microglia) compared to the control ones (59% of microglia) (Fig. 4I–K). Consistent with high proliferation of cortical microglia at P3 (Figs. 1C,E), we found three clusters of cycling microglia, representing close to 30% of total microglia, in both control and mutant conditions (Fig. 4I–L). Interestingly, we identified a cluster of immature microglia characterized by expression of genes such as *MEV2a7*, *Ccr1*, *Apoe*, *Cybb*, *Ms4a7*, and *MEV2a6c*, initially found enriched in the embryonic brain (Hammond et al, 2019; Ostrem et al, 2024). In *Emx1*$^{cre}$; *Csf1*$^{fl/fl}$ mutants, this cluster was abnormally enriched at P3 (2% in controls and 11% in mutants) (Figs. 4J–L and EV4). The distribution of microglia within clusters was significantly different in controls and mutants ($\chi^2$ = 159.76 et p value = 1.64 × 10$^{-33}$) (Fig. 4K). Finally, an ATM cluster characterized by *Spp1* (OPN), *Gpnmb*, *Igf1* and *Lgals3* expression was found in similar proportions in control and mutant mice (Fig. 4I–K), suggesting that the absence of *Csf1* did not affect the induction of this specific state.

Taken together, our data confirm that the action of cortical *Csf1* is transient, consistent with the presence of other potential sources of this ligand. Alongside, these findings suggest that although early repopulating microglia follow a layer-specific distribution pattern similar to controls, they display features of immature microglia but can nonetheless adopt ATM features in response to the local brain environment.

## ATM proliferation at hotspots is regulated by intrinsic microglial *Csf1*

Our data indicate that *Csf1* from cortical progenitors, and to a lesser extent from neurons, is locally essential for embryonic microglia. In addition, the scRNA-seq dataset highlighted that *Csf1* is also expressed by some microglia (Fig. EV2A)(Di Bella et al, 2021), in particular by pre- and

postnatal ATM (Benmamar-Badel et al, 2020; Hammond et al, 2019; La Manno et al, 2021; Lawrence et al, 2024) as found in the dataset from Hammond and colleagues (Hammond et al, 2019; Lawrence et al, 2024).

Since pre- and post- natal ATM are detected at the E14.5 CSA (Lawrence et al, 2024) and P7 EDWM (Hagemeyer et al, 2017; Hammond et al, 2019; Li et al, 2019; Wlodarczyk et al, 2017), two hotspots of proliferation, we wondered whether *Csf1* is expressed in these microglia and whether it could participate to their sustained proliferation. In both hotspots, microglia expressed core ATM makers including Osteopontin (coded by *Spp1*), Galectin3 (Gal3 encoded by *Lgals3*) and GPNMB (Fig. 5A–D). Notably, around 60% of microglia accumulating in the CSA, and about 80% in the EDWM, were OPN-positive (Fig. 5E–J). We hence used OPN as a generic marker for these cells to assess whether ATM proliferates more than neighboring microglia within hotspots, focusing on the CSA at E14.5 and the EDWM at P7. EdU injections showed that OPN-positive microglia were more proliferative than OPN-negative microglia in both hotspots (Fig. 5G,J), revealing a specific enhanced proliferation of ATM.

Transcriptomic analyses indicate that *Csf1* expression is restricted to ATM (Fig. 6A). We used RNAscope to confirm that ameboid ATM microglia at the E14.5 CSA and in the P7 EDWM expressed *Csf1*, in contrast to adjacent microglia that lacked *Csf1* transcripts (Fig. 6B–E). To investigate the role of ATM-intrinsic *Csf1*, we used *Cx3cr1*$^{creER}$;*Csf1*$^{fl/fl}$ mice to inactivate *Csf1* in microglia and macrophages upon tamoxifen administration at either E12.5 (to target the CSA) or P3 (to target the EDWM)(Fig. 6B–E). Using RNAscope, we showed that these procedures triggered a partial inactivation of *Csf1* at both the CSA and EDWM: quantification of *Csf1* signals in controls and mutants revealed a 47% decrease in expression at E14.5 within CSA microglia and a 43% decrease at the EDWM (Fig. 6C,E). Under these conditions, microglia still accumulated at the CSA and EDWM, expressing core ATM markers (Fig. EV5A–D), and the proportion of OPN-positive microglia remained unchanged (Fig. 6F–H). These data suggest that intrinsic microglial *Csf1* is not required for the induction of the ATM profile. Despite partial recombination, we assessed microglial proliferation using EdU labeling and found a reduction

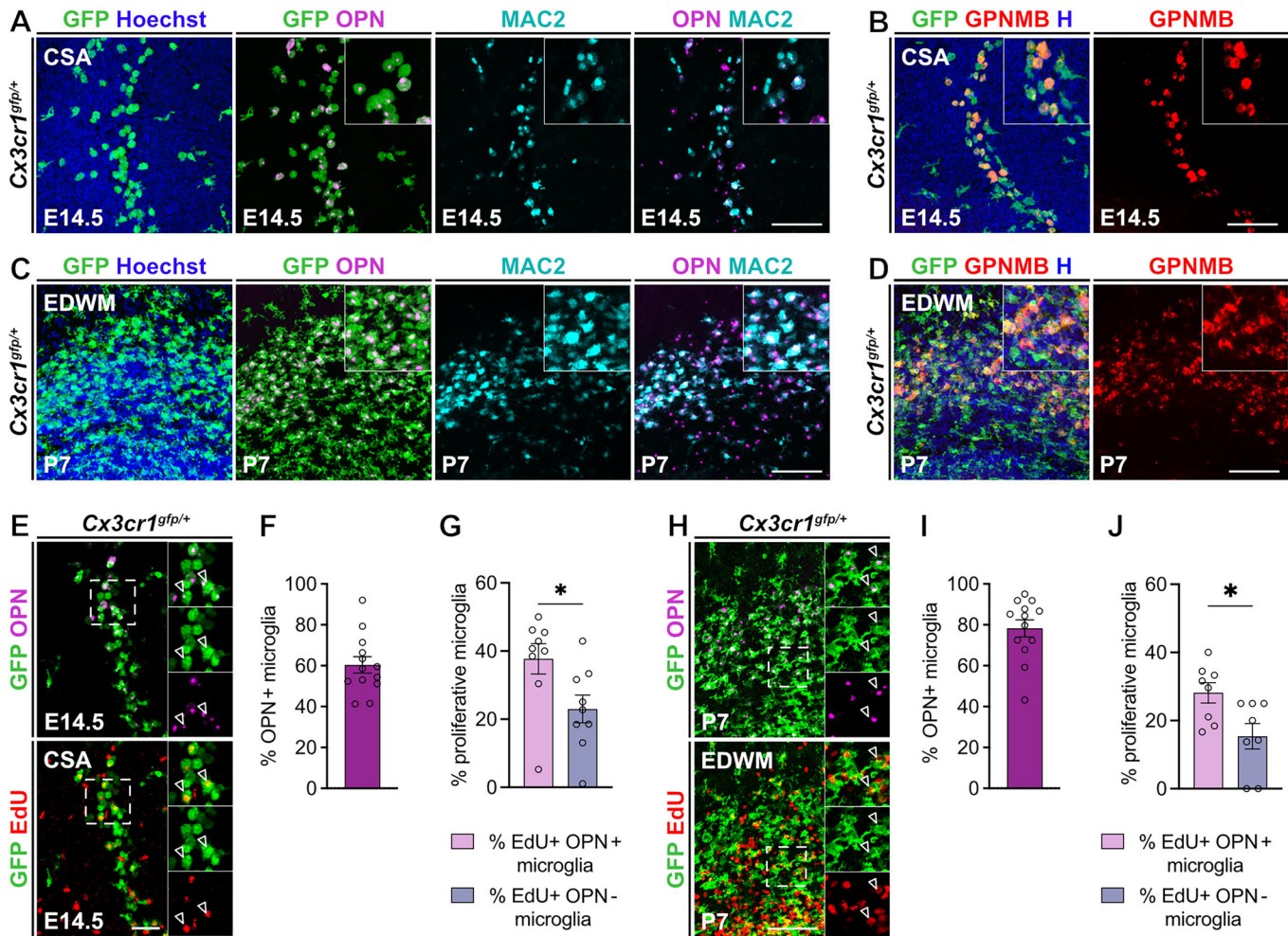

**Figure 5. ATM cells in transient development hotspots are preferentially proliferating.**

(A) Immunolabeling of coronal brain sections from E14.5 *Cx3cr1*[gfp/+] embryos showing co-expression of GFP, MAC2 and OPN at the CSA (*n* = 3). Close-ups are 75 μm wide. (B) Immunolabeling of coronal brain sections from E14.5 *Cx3cr1*[gfp/+] embryos showing co-expression of GFP and GPNMB at the CSA (*n* = 4). (C) Immunolabeling of coronal brain sections from P7 *Cx3cr1*[gfp/+] pups showing co-expression of GFP, MAC2 and OPN at the EDWM (*n* = 3). (D) Immunolabeling of coronal brain sections from P7 *Cx3cr1*[gfp/+] pups showing co-expression of GFP and GPNMB at the EDWM (*n* = 3). (E) Immunolabeling of coronal brain sections from E14.5 *Cx3cr1*[gfp/+] embryos showing co-expression of GFP and EdU (open arrowheads) at the CSA. (F) Percentage of OPN-positive ameboid microglia at the CSA from E14.5 embryos (*n* = 13; from at least two distinct litters). (G) Percentage of EdU-positive microglia at the CSA at E14.5 amongst OPN-positive or OPN-negative ameboid microglia (*n* = 9; from one litter) (*P* = 0.0197). (H) Immunolabeling of coronal brain sections from P7 *Cx3cr1*[gfp/+] embryos showing co-expression of GFP and EdU (open arrowheads) at the EDWM. (I) Percentage of OPN-positive ameboid microglia at the EDWM from P7 pups (*n* = 13; from at least two distinct litters). (J) Percentage of EdU-positive ameboid microglia at the EDWM at P7 amongst OPN-positive or OPN-negative ameboid microglia (*n* = 8; from two distinct litters) (*P* = 0.0272). Data were presented as mean ± SEM. Two-sided unpaired Mann–Whitney test were performed to assess differences (G, J). *P < 0.05. Scale bars: 100 μm (A–D, H); 50 μm (E). Close-ups are 100 μm (A–D) and 75 μm (E, H) wide. CSA cortico-striato-amygdalar boundary, EDWM early-dorsal white matter, H Hoechst. See Figs. EV2 and EV5. Source data are available online for this figure.

in the proportion of EdU-positive microglia at both the CSA and EDWM (Fig. 6I–K). Importantly, in this context, microglial density or proliferation in the somatosensory cortex was unchanged, consistent with a specific role of *Csf1* in ATM microglia (Fig. EV5E–G). Finally, to assess whether reducing microglial *Csf1* might ultimately affect the density of the ATM hotspot, we attempted to increase the efficiency of recombination of the *Csf1* locus by injecting tamoxifen at both P1 and P3 (Fig. EV5H–K). Quantification of *Csf1* signals in controls and mutants within EDWM microglia revealed a 72% decrease in expression (Fig. EV5H,J). We further observed a significant reduction of the density of microglia and ATM, as identified by OPN staining in the

EDWM (Fig. EV5I,K), supporting a role for microglial *Csf1* in the regulation of hotspots.

Overall, our data show that in addition to extrinsic neural *Csf1*, which is needed for microglial survival and proliferation, intrinsic expression of *Csf1* in ATM, accumulating at transient hotspots, contributes to their enhanced proliferation. The pattern of microglial colonization is thus finely regulated by distinct sources of *Csf1*, of both neural and microglial origin, which act jointly and locally. Taken together, our results shed new light on how microglia distribute and proliferate within the brain, as well as on the modes of action of *Csf1*, which have major implications for understanding how microglia colonize the brain in health and disease.

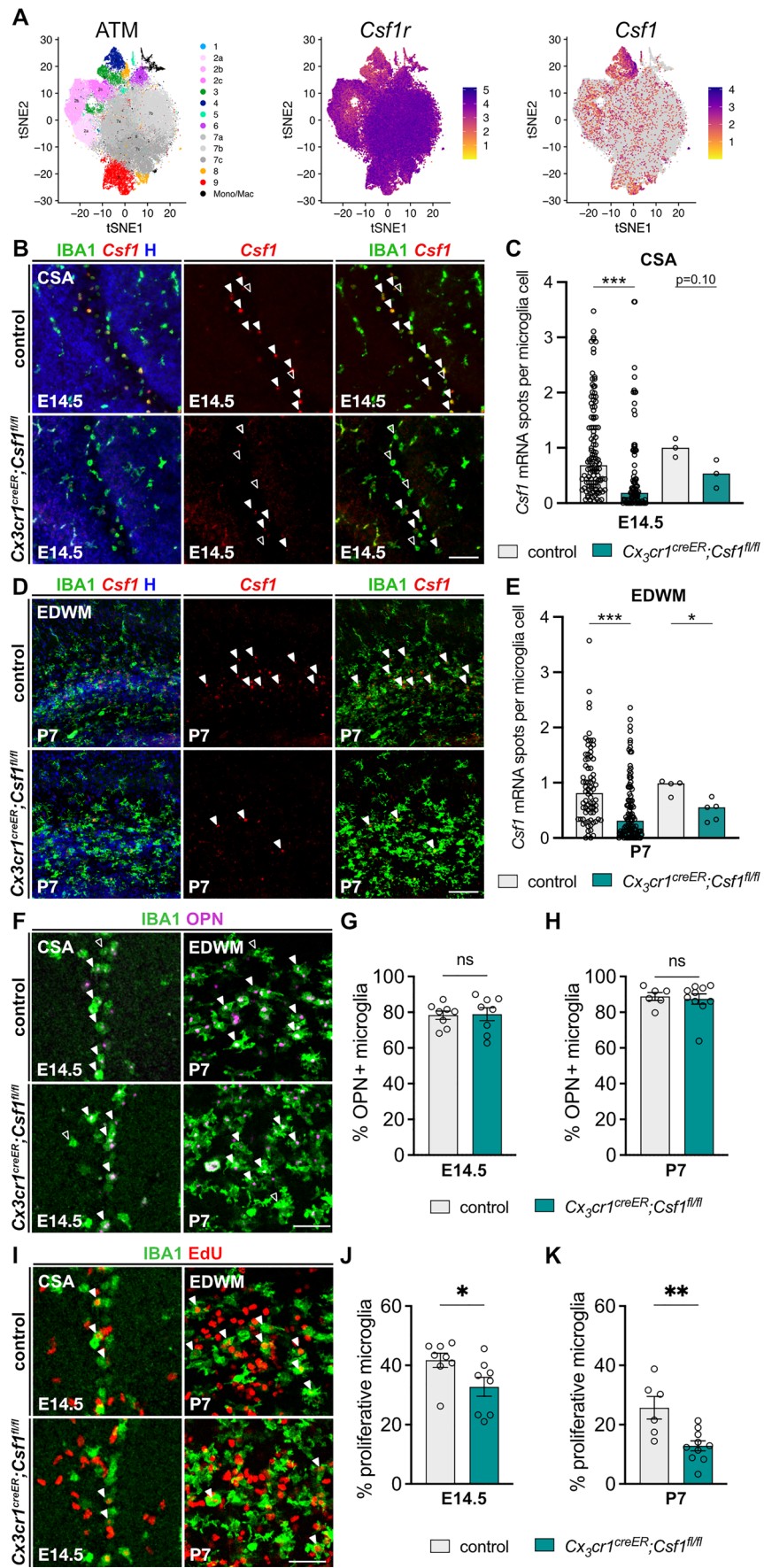

**Figure 6.    Intrinsic microglial *Csf1* expression regulates proliferation of ATM in transient developmental hotspots.**

(A) Reproduced tSNE plot of the 76,149 cells characterized by scRNA-seq in the dataset from (Hammond et al, 2019; Lawrence et al, 2024) and projections of the respective expression of *Csf1r* and *Csf1*. *Csf1* expression in particular overlaps with cluster 4 (ATM). (B) RNAscope experiments on coronal brain sections from E14.5 control and *Cx3cr1^{creER/+};Csf1^{fl/fl}* embryos showing *Csf1* expression in IBA1-positive ameboid microglia at the CSA (solid arrowheads). More microglia in *Cx3cr1^{creER/+};Csf1^{fl/fl}* embryos do not express *Csf1* (open arrowheads) after tamoxifen administration at E12.5. (C) Normalized quantification of RNAscope experiments on coronal brain sections from E14.5 control and *Cx3cr1^{creER/+};Csf1^{fl/fl}* embryos showing decreased *Csf1* expression in mutant mice. Left column shows all quantified cells and right column the mean expression per animal ($n_{control} = 3$; $n_{mutant} = 3$; from two distinct litters) (all cells, left: $P < 0.0001$; mean per animal, right: $P = 0.10$). (D) RNAscope experiments on coronal brain sections from P7 control and *Cx3cr1^{creER/+};Csf1^{fl/fl}* pups showing *Csf1* expression in IBA1-positive ameboid microglia at the CSA (solid arrowheads) after tamoxifen administration at P3. (E) Normalized quantification of RNAscope experiments on coronal brain sections from P7 control and *Cx3cr1^{creER/+};Csf1^{fl/fl}* embryos showing decreased *Csf1* expression in mutant mice. Left column shows all quantified cells and right column the mean expression per animal ($n_{control} = 4$; $n_{mutant} = 5$; from two distinct litters) (all cells, left: $P < 0.0001$; mean per animal, right: $P = 0.0317$). (F) Immunolabeling of coronal brain sections from control and *Cx3cr1^{creER/+};Csf1^{fl/fl}* mice showing co-expression of IBA1 and OPN in ameboid microglia within the CSA accumulation at E14.5 (tamoxifen administered at E12.5) or within the EDWM accumulation at P7 (tamoxifen administered at P3). Solid arrowheads point at microglia that co-expressed IBA1 and OPN, while open arrowheads point at microglia that did not co-express OPN. (G) Proportion of OPN-positive microglia at the CSA at E14.5 ($n_{control} = 8$; $n_{mutant} = 8$; from two distinct litters). (H) Proportion of OPN-positive microglia at the EDWM at P7 ($n_{control} = 6$; $n_{mutant} = 10$; from two distinct litters). (I) Immunolabeling of coronal brain sections from control and *Cx3cr1^{creER/+};Csf1^{fl/fl}* mice showing co-expression of IBA1 and EdU in ameboid microglia within the CSA accumulation at E14.5 (tamoxifen administered at E12.5) or within the EDWM accumulation at P7 (tamoxifen administered at P3). Solid arrowheads indicate proliferating (EdU-positive) microglia. (J) Proportion of proliferating (EdU positive) microglia at the CSA at E14.5 ($n_{control} = 8$; $n_{mutant} = 8$; from two distinct litters) ($P = 0.0207$). (K) Proportion of proliferating (EdU positive) microglia at the EDWM at P7 ($n_{control} = 6$; $n_{mutant} = 10$; from two distinct litters) ($P = 0.0047$). Data were presented as mean ± SEM. Two-sided unpaired Mann–Whitney test were performed to assess differences (C, E, G, H, J, K). ns not significant, $*P < 0.05$, $**P < 0.01$, $***P < 0.001$. Scale bars: 100 μm (B, D); 50 μm (F, I). CSA cortico-striato-amygdalar boundary, EDWM early-dorsal white matter, H Hoechst. See Figs. EV2 and EV5. Source data are available online for this figure.

# Discussion

Microglia adopt a stereotyped and heterogeneous pattern of distribution during early brain development, which is important for their functions. Here, we show that this pattern is regulated by temporally distinct waves of proliferation and specific proliferative hotspots where microglia accumulate in an ATM state. While the production of *Csf1* by the local neural environment is required for the proliferation and maintenance of microglial cells, ATM self-sustain their proliferation by producing *Csf1*, highlighting a specific property of this state. Collectively, our study reveals a remarkable crosstalk between *Csf1* sources, microglial niches, and cellular states in the regulation of their distribution and colonization.

Microglial colonization begins early in development, concurrent with the generation of the first neurons, as shown by immunostaining (Alliot et al, 1999; Barry-Carroll et al, 2023; Dalmau et al, 2003; Menassa and Gomez-Nicola, 2018; Menassa et al, 2022; Swinnen et al, 2013; Verney et al, 2010), scRNA-seq studies (Hammond et al, 2019; La Manno et al, 2021; Li et al, 2019; Ostrem et al, 2024) and lineage- and clonal- tracing (Barry-Carroll et al, 2023; Ratz et al, 2022) studies. This is consistent with the microglial population being largely generated from a restricted pool of progenitors that settles in the parenchyma during embryogenesis. By performing a longitudinal study of murine microglial proliferation in different regions of the forebrain during development, we here confirmed and extended previous findings in mouse (Barry-Caroll et al, 2023; Nikodemova et al, 2015; Ostrem et al, 2024; Swinnen et al, 2013) and human (Menassa et al, 2022), showing dynamic phases of microglial expansion. We identified an early phase of intense global proliferation, followed by a period of reduced proliferation between E16.5 and birth, which correlated with decreased microglial densities across regions. This supports the notion that, during this phase of brain expansion, the entry of microglia into the brain parenchyma is scarce or absent. We also observed a second postnatal phase of proliferation, which exhibited heterogeneous timing and intensity between forebrain regions. After the second postnatal week, proliferation was greatly reduced and microglial densities progressively reached adult levels, as previously described

(Hope et al, 2020; Menassa et al, 2022; Nikodemova et al, 2015). In parallel, we showed that transient hotspots of ameboid microglia at the CSA and in the cortical white matter (EDWM)(Hristova et al, 2010; Kershman, 1939; Verney et al, 2010) display sustained proliferation. By studying microglial dynamics in the cortex and the CSA at E14.5, when microglia are particularly mobile, our study overall supports the notion that microglial expansion is mainly driven by local proliferation rather than long-range migration.

Although microglial proliferation is independent of surrounding cell death or proliferation in the basal ganglia (Hope et al, 2020), little is known about the signals inducing the different proliferative waves and the regional differences that we and others have highlighted (Dalmau et al, 2003; Hope et al, 2020; Lawrence et al, 2024; Squarzoni et al, 2014). It is noteworthy that early perturbations of cortical neuronal activity, did not have any discernable effect on microglial numbers or distribution—even though microglia respond to patterns of neuronal activity in other contexts (Iaccarino et al, 2016). This is in sharp contrast with a recently reported role of cortical activity in microglial proliferation during the second postnatal week (Kumaraguru et al, 2025), highlighting that microglial colonization is regulated by a variety of processes that can differ across stages and regions. Instead, we confirmed the assertion made both through in vitro (Stanley and Chitu, 2014) and in vivo overexpression studies (De et al, 2014), that CSF-1 is a key driver of microglial proliferation. Here, we revealed that the sources of neocortical CSF-1 are mainly progenitors, and, to a lesser extent, immature neurons, in agreement with other studies (Di Bella et al, 2021). We found that CSF-1 acted very locally, suggesting that circulating CSF-1, which is found in the blood throughout life (Stanley and Chitu, 2014), is not sufficient for microglial maintenance in the embryonic neocortex.

In the *Emx1^{cre};  Csf1^{fl/fl}* model, microglia were strikingly absent from the neocortex at E14.5. This could either reflect their incapacity to reach the parenchyma due to the lack of a CSF-1 chemoattractant effect, or their death in the absence of CSF-1. Nevertheless, a few progenitors did reach the brain parenchyma, as we found some microglial cells in the neocortex at E12.5. Together with the finding that there is only a mild impact on microglial

densities in *Nex^cre^; Csf1^fl/fl^*, our analysis suggests that ventricular cortical progenitors expressing *Csf1* provide a gateway for microglial colonization of the neocortex. Interestingly, and consistent with previous findings (Easley-Neal et al, 2019; Kana et al, 2019; Nandi et al, 2012), CSF-1 is only transiently essential for microglial colonization, as evidenced by their drastic depletion in *Emx1^cre^; Csf1^fl/fl^* mice. In parallel, this model suggests that neural IL34 is not required for embryonic microglial proliferation and survival, in agreement with the absence of effect observed at birth in full *Il34* knockout (Greter et al, 2012), and with the lack of *Il34* expression by neural cells and microglia during brain development (La Manno et al, 2021). Nevertheless, the postnatal expression of *Il34* by cortical cells along development, which is crucial for postnatal microglial density and maturation (Devlin et al, 2025; Greter et al, 2012; Wang et al, 2012), could be important for microglial repopulation in the absence of *Csf1* in *Emx1^cre^; Csf1^fl/fl^* mice. The ameboid morphology and decreased P2RY12 expression of microglia during the early phase of repopulation, as well as their enriched "immature" profile highlighted through scRNA-seq, is reminiscent of repopulating microglia following CSF1R inhibition (Elmore et al, 2014; Zhan et al, 2020) and consistent with a stepwise program of microglial maturation (Matcovitch-Natan et al, 2016). Additionally, other sources of both CSF-1 and IL34, for instance, pericytes or endothelial cells, could also contribute to microglial maintenance. Furthermore, we confirmed and extended previous findings highlighting a conserved temporal and cell-type-specific expression pattern of *Csf1*, *Il34*, and *Csf1r* in both mouse and human brains, reinforcing the relevance of our functional analyses in mice for understanding human brain development. Our data not only reveal much-needed insight into the processes underlying microglial colonization, but also highlight a novel model for specific, local, and transient microglial depletion. Furthermore, this study provides proof-of-concept for targeting local niches of microglia, allowing us to dissect their specific functions, something that has long been lacking in the field (Bridlance and Thion, 2023). This model will enable us to assess how early changes in localization could affect the long-term development of the brain, from an anatomic or behavioral point of view. Indeed, microglial density and colonization pattern is related to the established developmental functions of microglia (Thion and Garel, 2017; Thion and Garel, 2020; Thion et al, 2018a).

Last, in addition to neural *Csf1*, we showed that intrinsic microglial *Csf1*, restricted to the ATM profile found in developmental hotspots, is important for their sustained proliferation. This surprising finding suggests that the ATM-like state could allow microglia to be somewhat independent of neural CSF-1, as well as to increase their proliferation and consequently their density. In both hotspots, microglia exert specific functions crucial for proper brain development, including maintenance of tissue integrity at the embryonic CSA (Lawrence et al, 2024), as well as gliogenesis and myelin growth in the EDWM (Hammond et al, 2019; Li et al, 2019; McNamara et al, 2023; Nemes-Baran et al, 2020). Consistent with the importance of having a high density of microglia at these hotspots to maintain tissue integrity (Lawrence et al, 2024), our analyses of microglial dynamics indicated that ATM accumulations are linked to the combined local proliferation and the absence of migration and dispersion. Finally, Disease-Associated Microglia (DAM), which share many characteristics with ATM, including *Csf1* expression (Bisht et al, 2016; Catale and Garel, 2025; Keren-

Shaul et al, 2017; Krasemann et al, 2017; Marschallinger et al, 2020; Masuda et al, 2019; Safaiyan et al, 2016; Sala Frigerio et al, 2019; Silvin et al, 2022), could potentially employ similar self-enhanced *Csf1*-dependent proliferation to expand upon neurodegenerative conditions. Further studies are warranted to examine the cell-autonomous roles of *Csf1* in microglial proliferation and functions across developmental stages and states.

Our findings support a model in which microglial colonization of the developing brain occurs in at least two distinct phases. During the early embryonic period, microglial proliferation and migration rely predominantly on CSF-1 signaling. This phase ensures the initial seeding and expansion of microglia throughout the brain parenchyma, largely independent of neuronal activity. Postnatally, as neural circuits mature and neuronal firing patterns emerge, microglia adapt their density and morphology in response to local activity cues (Kumaraguru et al, 2025), contributing to synaptic pruning and circuit refinement. This shift in microglial regulation from a growth factor-driven to an activity-dependent mechanism likely reflects the changing demands of the maturing brain environment.

Overall, our study highlights waves and hotspots of proliferation driving the stereotyped pattern of microglial distribution as they colonize the brain, which is essential for their developmental functions. While the CSF1R signaling has long been recognized as essential for microglial survival and development, our work furthermore reveals that such an early microglial pattern of distribution is finely regulated by distinct sources of CSF-1, of neural and microglial origin, that both act at short range, providing key insights into how important immune cells colonize the brain during a critical developmental time window.

### Limitations of the study

We characterized microglial proliferation and densities in different regions of the brain through development, and defined the implications of local expression of CSF-1. A limitation of our study was the inability to fully recombine the *Csf1^fl^* allele using the *Cx3cr1^creER^* line, which prevented us from fully testing the contribution of cell-autonomous sources of *Csf1* in the expansion of ATM hotspots, as well as their potential functions. Nonetheless, our findings shed light on how microglia proliferate and distribute to colonize the brain as well as on their focal and cell-type dependency on *Csf1*.

## Methods

**Reagents and tools table**

| Reagent/resource | Reference or source | Identifier or catalog number |
|---|---|---|
| **Experimental models** | | |
| Mouse : C57BL/6J | Charles River | 632C57BL/6 J |
| Mouse : Cx3cr1^gfp/+^ | Jung et al, 2000 | JAX:005582 |
| Mouse : Cx3cr1^creER^ | Yona et al, 2013 | JAX:020940 |
| Mouse : Emx1^cre/+^ | Gorski et al, 2002 | JAX:005628 |
| Mouse : Nex^cre/+^ | Goebbels et al | N/A |
| Mouse : Il34^fl/fl^ | Greter et al | N/A |

| Reagent/resource | Reference or source | Identifier or catalog number |
|---|---|---|
| Mouse : *Csf1*^fl/fl | Harris et al | N/A |
| Mouse : *R26*^loxP-stop-loxPtdTomato | Madisen et al | N/A |
| Mouse : *R26*^Kir2.1−mCherry/+ | Moreno-Juan et al | N/A |
| Mouse : *Nestin*^cre/+ | Tronche et al, 1999 | JAX:003771 |
| **Recombinant DNA** | | |
| **Antibodies** | | |
| Chicken anti-GFP | Aves Labs | Cat# GFP-1020 |
| Chicken anti-IBA1 | Synaptic Systems | Cat# 234 009 |
| Goat anti-mouse Osteoactivin | R and D Systems | Cat# AF2330 |
| Goat anti-OPN | R and D Systems | Cat# AF808 |
| Guinea pig anti-VGlut2 | Millipore | Cat# AB2251-I |
| Rabbit anti-IBA1 | FUJIFILM Wako Shibayagi | Cat# 019-19741 |
| Rabbit anti-KI67 | Abcam | Cat# ab15580 |
| Rabbit anti-P2RY12 | AnaSpec | Cat# 55043 A |
| Rat anti-CD206 | Bio-Rad | Cat# MCA2235 |
| Rat anti-GAL3 | Cedarlane | Cat# CL8942AP |
| Rat anti-Lyve1 | Thermo Fisher Scientific | Cat# 14-0443-82 |
| Donkey anti-chicken Alexa 10 Fluor® 488-conjugated | Jackson ImmunoResearch Labs | Cat# 703-545-155 |
| Donkey anti-goat Alexa 10 Fluor® 488-conjugated | Jackson ImmunoResearch Labs | Cat# 705-545-147 |
| Donkey anti-rat Alexa 10 Fluor® 488-conjugated | Jackson ImmunoResearch Labs | Cat# 712-545-150 |
| Donkey anti-rabbit Alexa 10 Fluor® 488-conjugated | Jackson ImmunoResearch Labs | Cat# 711-545-152 |
| Donkey anti-goat Cy3-conjugated | Jackson ImmunoResearch Labs | Cat# 705-165-147 |
| Donkey anti-rabbit Cy3-conjugated | Jackson ImmunoResearch Labs | Cat# 711-165-152 |
| Donkey anti-rat Cy3-conjugated | Jackson ImmunoResearch Labs | Cat# 712-165-150 |
| Donkey anti-guinea pig Cy3- conjugated | Jackson ImmunoResearch Labs | Cat# 706 165 148 |
| Donkey anti-goat Alexa 10 Fluor® 647-conjugated | Jackson ImmunoResearch Labs | Cat# 705-605-147 |
| Donkey anti-goat Cy5-conjugated | Jackson ImmunoResearch Labs | Cat# 705-175-147 |
| Donkey anti-rat Cy5-conjugated | Jackson ImmunoResearch Labs | Cat# 712-175-150 |

| Reagent/resource | Reference or source | Identifier or catalog number |
|---|---|---|
| **Oligonucleotides and other sequence-based reagents** | | |
| RNAscope probe Csf1 | ACD | Cat# 122261-C3 |
| RNAscope probe Csf1 | ACD | Cat# 315621-C3 |
| **Chemicals, enzymes, and other reagents** | | |
| BD RhapsodyTM Whole Transcriptome Analysis (WTA) Amplification Kit | BD Biosciences | Cat# 633801 |
| BDTM Ms Single Cell Sample Multiplexing Kit | BD Biosciences | Cat# 633793 |
| CD45 microbeads from Miltenyi | BD Biosciences | Cat# 130-052-301 |
| Click-iT EdU Alexa Fluor 647 imaging kit | Invitrogen | Cat# C10340 |
| EdU | Thermo Fisher | Cat# A10044 |
| Gelatin | VWR chemical | Cat# 24350.262 |
| Glucose | Sigma-Aldrich | Cat# G7021 |
| Hoechst | Sigma-Aldrich | Cat# 33342 |
| L-15 Medium without phenol | Gibco | Cat# 21083-027 |
| Low-Melting Agarose | Invitrogen | Cat# 16520 |
| Opal 570 | Akoya Biosciences | Cat# FP1488001KT |
| Opal 690 | Akoya Biosciences | Cat# FP1497001KT |
| Paraformaldehyde (PFA) | Sigma-Aldrich | Cat# P6148 |
| Phosphate Buffer Saline (PBS) | Eurobio | Cat# GAUPBS00-07 |
| RNAscope Fluorescent Multiplex Reagent Kit | Advanced Cell Diagnostics-Biotechne | Cat# 323100 |
| Transcriptome Analysis (WTA) Reagent Kit | BD Biosciences | Cat# 665915 |
| Triton X-100 | Sigma-Aldrich | Cat# 1086031000 |
| Vectashield | Vector Laboratories | Cat# H-1000 |
| **Software** | | |
| FIJI (ImageJ) 1.50 | National Institute of Health | https://fiji.sc/ https://imagej.nih.gov/ij/index.html |
| GraphPad Prism 9.5 | GraphPad Software | https://www.graphpad.com/features |
| Illumina HiSeq 4000 system | Ilumina | https://www.illumina.com/systems/sequencing-platforms/hiseq-3000-4000.html |
| QuPath software | National Institute of Health | https://qupath.github.io |
| R package: Seurat 4.3.0.1 | N/A | https://satijalab.org/seurat/get_started.html |
| R software 4.2.2 | GNU Project | https://www.r-project.org/ |

| Reagent/resource | Reference or source | Identifier or catalog number |
|---|---|---|
| Rhapsody analysis pipeline | BD Biosciences | https://www.bdbiosciences.com |
| **Other** | | |
| Leica DMi8 fluorescence microscope | Leica | https://www.leica-microsystems.com/ |
| Leica TCS-SP5 confocal microscope | Leica | https://www.leica-microsystems.com/ |
| Leica TCS-SP8 confocal microscope | Leica | https://www.leica-microsystems.com/ |
| microscope ECHO Revolve | Echo | https://discover-echo.com/revolve/ |
| Yokogawa CSUX1-A1 confocal spinning disk microscope | Yokogawa | https://www.yokogawa.com |

## Methods and protocols

### Mouse lines

All mice were housed under a 12 h light-dark cycle at constant temperature (20–24 °C) and humidity (55 ± 10%) with ad libitum access to food and water. Wild-type littermates were used as controls for mutant mice. We used both males and females throughout our experiments, and detailed information on male/female comparison and pooling can be found in the section on statistical analysis. Mice were genotyped by tail biopsy and PCR using primers specific for the different alleles as defined by the provider or initial publications. The day of vaginal plug formation was considered E0.5, and the birth date was considered postnatal day 0 (P0). Mice were handled in accordance with European regulations and those of the local ethics committee (Reference numbers #24597 and #52515). $Csf1^{fl/fl}$ (Harris et al, 2012), $Cx3cr1^{gfp/+}$ (JAX:005582)(Jung et al, 2000), $Cx3cr1^{creER}$ (JAX:020940) (Yona et al, 2013), $Emx1^{cre/+}$ (JAX:005628)(Gorski et al, 2002), $Il34^{fl/fl}$ (Greter et al, 2012), $Nestin^{cre/+}$ (Tronche et al, 1999), $Nex^{cre/+}$ (Goebbels et al, 2006), and $R26^{loxP-stop-loxPtdTomato}$ ($R26^{mT/mT}$; JAX:007909)(Madisen et al, 2010) mice were maintained on a C57BL/6 J background, while $R26^{Kir2.1-mCherry/+}$ ($R26^{Kir2.1/+}$)(Moreno-Juan et al, 2017) were maintained on an ICR/CD-1/C57Bl/6J mixed background.

To overexpress the Kir2.1 channel in cortical progenitors and excitatory neurons, $Emx1^{cre/+}$ mice were crossed with the $R26^{Kir2.1-mCherry/+}$ line. To conditionally inactivate $Csf1$ in monocytes and macrophages, $Csf1^{fl/fl}$ mice were backcrossed onto $Cx3cr1^{creER}$ to obtain $Cx3cr1^{creER/+}$;$Csf1^{fl/fl}$ mice. Tamoxifen induction of CreER-driven recombination was performed either at E12.5 by oral gavage of pregnant dams with 300 μl of tamoxifen (Sigma) dissolved in corn oil at a final concentration of 2 mg/10 g animal; or at P1 and/or P3 by subcutaneous injection of 10 μl/g animal of tamoxifen dissolved in corn oil at a final concentration of 2 mg/10 g animal. To conditionally inactivate $Csf1$ in cortical progenitors and excitatory neurons, or in cortical neurons only, $Csf1^{fl/fl}$ mice were backcrossed to $Emx1^{cre/+}$ or $Nex^{cre/+}$ mice, respectively. Backcrossing $Csf1^{fl/fl}$ mice to the $R26^{mT/mT}$ line allowed us to visualize $Emx1^{cre/+}$ driven recombination pattern at E10.5 and E14.5. Last, to conditionally inactivate $Il34$ in cortical progenitors and neurons, $Il34^{fl/fl}$ mice were backcrossed to $Emx1^{cre/+}$ mice.

### Sensory deprivation

Whisker plucking of pups was performed bilaterally from P1 to P7 as previously described (Genescu et al, 2022). The infraorbital nerve (ION) lesion was performed unilaterally at P1, as previously described (Frangeul et al, 2014; White et al, 1990). The efficiency of the manipulation was assessed by anti-vGlut2 immunostaining to confirm the absence of barrels in the contralateral somatosensory cortex.

### Tissue preparation and immunohistochemistry

For experiments at embryonic stages, pregnant females were sacrificed by cervical dislocation, embryos were collected, and their brains were immediately dissected, except for E12.5 embryos in which the whole head was kept intact. For experiments at postnatal stages, pups were anesthetized and transcardially perfused with 4% paraformaldehyde (PFA) in PBS, preceding brain dissection. Brains were then fixed in 4% PFA at 4 °C for 12–18 h at room temperature (4 h for E12.5 embryos). Brains were cut into coronal free-floating sections of 80 μm (embryonic and P0 brains) or 60 μm (postnatal brains) thickness. Slices were first incubated for 1 h at room temperature (RT) in 0.2% Triton X-100, 0.2% Gelatin in PBS (blocking solution), and then incubated in the same blocking solution with the following primary antibodies overnight at 4 °C: rat anti-CD206 (1/200; Bio-Rad Cat# MCA2235, RRID:AB_324622), chicken anti-GFP (1/1000; Aves Labs Cat# GFP-1020, RRID:AB_10000240), rabbit anti-IBA1 (1/500; FUJIFILM Wako Shibayagi Cat# 019-19741, RRID:AB_839504), chicken anti-IBA1 (1/400; Synaptic Systems Cat# 234 009, RRID:AB_2891282), rabbit anti-KI67 (1/200; Abcam Cat# ab15580, RRID:AB_443209), rat anti-Lyve1 (ALY7) (1/200; Thermo Fisher Scientific Cat# 14-0443-82, RRID:AB_1633414), rat anti-GAL3 (MAC2) (1/1000; Cedarlane Cat# CL8942AP; RRID:AB_10060357), goat anti-OPN (1/400; R and D Systems Cat# AF808, RRID:AB_2194992), goat anti-mouse Osteoactivin (GPNMB) (1/200; R and D Systems Cat# AF2330; RRID:AB_2112934), rabbit anti-P2RY12 (1/500; AnaSpec; EGT Group Cat# 55043 A, RRID:AB_2298886), and anti-VGlut2 (1/2000; Millipore Cat# AB2251-I, RRID:AB_2665454). Sections were rinsed in PBS 0.1% Triton X-100 and incubated from 2 to 18 h at 4 °C with secondary antibodies (1/400 in PBS, Jackson ImmunoResearch Labs): Alexa 10 Fluor® 488-conjugated donkey anti-chicken (Cat# 703-545-155, RRID:AB_2340375), Alexa 10 Fluor® 488-conjugated donkey anti-goat (Cat# 705-545-147, RRID:AB_2336933); Alexa 10 Fluor® 488-conjugated donkey anti-rat (Cat# 712-545-150, RRID:AB_2340683); Alexa 10 Fluor® 488-conjugated donkey rabbit (Cat# 711-545-152, RRID:AB_2313584); Cy3-conjugated donkey anti-goat (Cat# 705-165-147, RRID:AB_2307351), Cy3-conjugated donkey anti-rabbit (Cat# 711-165-152, RRID:AB_2307443), Cy3-conjugated donkey anti-rat (Cat# 712-165-150, RRID:AB_2340666), Alexa 10 Fluor® 647-conjugated donkey anti-goat (Cat# 705-605-147, RRID:AB_2340437), Cy5-conjugated donkey anti-goat (Cat# 705-175-147, RRID:AB_2340415), and Cy5-conjugated donkey anti-rat (Cat# 712-175-150, RRID:AB_2340671). Hoechst (1/1000; Sigma-Aldrich 33342) was used for fluorescent nuclear counterstaining, and Vectashield was used for mounting (Vector Labs).

### EdU administration and detection

To monitor microglial proliferation, 50 mg/kg body weight of EdU (Thermo Fisher Scientific) dissolved in filtered PBS was

administered to pregnant mice by intraperitoneal injection or to pups by subcutaneous injection. Mice were sacrificed 2 h after injections, to ensure efficient EdU incorporation into dividing cells while limiting the number of EdU-positive cells in the meantime. EdU incorporation was detected using the Click-iT EdU Alexa Fluor imaging kit (Thermo Fisher Scientific for Alexa Fluor 647 staining), with a protocol adapted from (Podgorny et al, 2018). Briefly, brains were processed as described above, and free-floating slices were first blocked and then incubated overnight at 4 °C with primary antibodies for immunostaining. After three washes in PBS, slices were permeabilized in 4% Triton X-100 in PBS for 1 h. Slices were again washed in PBS three times and incubated for 1 h with the Click-iT reaction cocktail, protected from light. After three washes in PBS, sections were incubated with secondary antibodies and Hoechst for 2 h at RT, washed three times in PBS and mounted as described above.

### Combined immunohistochemistry and ACD RNAscope fluorescent multiplex in situ hybridization

Brains were processed as described above, and the following steps were carried out according to the ACD RNAscope Fluorescent Multiplex Reagent Kit protocol (Cat# 323100), with some adaptations to enhance the signal-to-noise ratio as well as use free-floating sections.

Briefly, for the RNAscope experiment, after three washes in PBS, slices that will be later mounted were first delipidated in PBS 0.3% Triton X-100 for 2 h at RT and then rinsed 5 min with PBS 0.3% Triton TX-100 while floating slices were first delipidated in PBS 1% Triton X-100, 0.3% PVSA (Sigma 278424, PubChem SID 24856814) for 2 h at RT and then rinsed 5 min with PBS 0.2% Triton TX-100 0.3% PVSA. Then, all slices were washed two times for 5 min with PBS 0.5% Tween-20 before $H_2O_2$ treatment. Slices were then either mounted on Superfrost+ slides and left to dry for 1 h at room temperature, 1 h at 60 °C in a drying oven, and overnight at room temperature or the experiment was performed on floating slices (improved signal to noise) that were stored O/N at 4 °C in PBS PVSA 0,3%. Mounted slices were then treated with protease+ for 30 min at room temperature, while floating slices were not. All slices were hybridized with *Csf1R* (ACD # 428191-C2) or/and *Csf1* (ACD # 1222261-C3) according to the ACD RNAscope Fluorescent Multiplex Reagent Kit protocol (user manual document number UM 323100), and probes were revealed with either vivid TSA 570 (ACD # 323272) for *Csf1R* mRNAs or vivid TSA 650 (ACD # 323273) for *Csf1* mRNAs.

For combined immunostaining, when the slices were mounted to perform the RNAscope, immunostaining was performed before hybridization, with free-floating slices blocked before incubation overnight at 4 °C with a Rabbit anti-IBA1 antibody (WAKO Cat# 019-19741, RRID: AB2665520), and the secondary antibody incubated after hybridization. For our improved adapted protocol on free-floating sections, immunostaining was entirely performed after hybridization, as previously described. For all conditions, slices were counterstained with Hoechst and mounted on Super-frost plus slides in Fluoromount medium (IN VITROGEN #00-4958-02).

### Two-photon live imaging

*Cx3cr1*<sup>gfp/+</sup> pups were sacrificed by decapitation, then their brains were dissected and kept in an ice-cold solution of L-15 Medium without phenol red (Gibco) supplemented with 3% Glucose. For embryos, pregnant dams were sacrificed by cervical dislocation, embryos were removed, and their brains dissected as before. Brains were embedded in 3.5% low-melting agarose (Promega) in L-15 Glucose+ without phenol red solution, then 300-μm-thick coronal slices were cut on a vibratome. Slices were allowed to equilibrate in L-15 Glucose+ solution at 37 °C and bubbled with $O_2/CO_2$ (95% / 5%) for 1 h at 37 °C. They were then imaged using a custom-designed AOD-based multi-photon microscope (AODscope, IBENS Imaging Facility) over a period of up to 6 h, while being constantly perfused with L-15 Glucose+ at 37 °C, bubbled with 5% $CO_2$. The microscope laser was tuned to 900 nm. Fluorescence was detected after a 510–84 nm band-pass filter to acquire GFP signals. Starting and finishing more than 100 μm away from the surface of the slice, 100-μm-thick z-stacks were imaged every 30 s with a z-step of 1 μm. Individual slices were imaged for up to 6 h without loss of GFP signals from microglia or signs of microglial death. For time-lapse movie analysis, eventual drift in the three dimensions was first corrected using IMARIS software (Bitplane, Oxford). To track microglial movements and analyze their migratory speed and directions, the functionality "Spots" was used and further manually corrected. Potential BAMs were also manually removed from the analysis.

### Transcriptomic analysis in controls and Emx1<sup>cre/+</sup>; Csf1<sup>fl/fl</sup> mutants

The Rhapsody experiment was carried out following the manufacturer's (BD Biosciences) protocol. Controls and mutants were littermates. CD45+ cells were enriched using CD45 microbeads from Miltenyi (Cat# 130-052-301) and the Automacs using possel function. About 37,205 cells were captured and processed in a single run. Five sample tags were used (comprising three control mice and three mutant mice), pooled using the BD™ Ms Single Cell Sample Multiplexing Kit (Cat# 633793). The sample was processed according to the BD Rhapsody™ Whole Transcriptome Analysis (WTA) Amplification Kit (Cat# 633801) and the BD Rhapsody™ Whole Transcriptome Analysis (WTA) Reagent Kit (Cat# 665915). The library was then subjected to an indexed paired-end sequencing run of 2 × 151 cycles on an Illumina HiSeq 4000 system (Illumina, San Diego, CA, USA) with 20% PhiX spike in. Over 3753 million reads were sequenced.

Transcriptomics Fastq files were processed via the standard Rhapsody analysis pipeline (BD Biosciences) as per the manufacturer's recommendations. First, R1 and R2 reads were filtered for quality, dropping reads that were too short (fewer than 66 bases for R1 and 64 bases for R2) or had a base quality score of less than 20. R1 reads are annotated to identify cell label sequences and unique molecular identifiers (UMIs), and R2 reads were mapped to the respective reference sequences using Bowtie2 (Langmead and Salzberg, 2012). Finally, all passing R1 and R2 reads were combined and annotated to the respective molecules. For quality control of the reads, recursive substation error correction (RSEC) and distribution-based error correction (DBEC) were applied, which are manufacturer-developed algorithms correcting for PCR and sequencing errors. For determining putative cells (which will contain many more reads than noise cell labels), a filtering algorithm took the number of DBEC-corrected reads into account, calculating the minimum second derivative along with the cumulative reads as the cut-off point. Finally, the expression matrix was obtained from the DBEC-adjusted molecule counts in a

CSV format. A cell was determined as a singlet if the minimum read count of a single sample tag was above the threshold of 75%. A cell was classified as a multiplet if the cell exceeded the threshold for more than one sample tag. A cell that did not meet the threshold was labeled as undetermined. Both multiplets and undetermined cells were excluded from the analysis as described below.

RSEC-adjusted molecule count matrices from the seven bridges BD rhapsody alignment pipeline were loaded and analyzed using Seurat v4.3.0.1. Samples were demultiplexed using the Sample Tag Call matrix from the Seven Bridges BD Rhapsody alignment pipeline and metadata was manually assigned to each Tag. High-quality cells were filtered and selected based on <25% of RNA from mitochondrial genes >200 of expressed unique genes and >100 of UMI counts, leading to a total of 19,746 cells, of which 3853 were microglia (2560 control and 1293 mutant) (Appendix Fig. S3). Count matrix was normalized using the SCTransform function from Seurat (v4) (defaults parameters). This method applies a regularized negative binomial regression to stabilize variance and correct for sequencing depth (Hafemeister and Satija, 2019). PCA, UMAP dimension reduction, nearest neighbors' determination and Louvain clustering were performed using standard Seurat functions and parameters. The most stable resolution of clustering (SCT_snn_res.0.3) was chosen according to the clustree v0.5.0 package guidelines (Zappia and Oshlack, 2018). Differential gene expression analysis was performed with the "FindMarkers" Seurat function using the Wilcoxon signed-rank test (Hao et al, 2021), minimum log-fold change threshold = 0.25, minimum percentage expression = 0.25 and FDR-adjusted p-value (Hao et al, 2021).

### Image acquisition and quantification

Immunofluorescence images were acquired using a confocal microscope (Leica TCS-SP5 and SP8) with a 10X or 20X objective, or a fluorescence microscope (Leica DMi8). Microglial quantifications were performed at each stage and region on whole-stack projections (Standard Deviation function) using ImageJ software, with the Cell Counter plug-in. Microglial proliferation was assessed by automated segmentation of the Ki67 channel (or alternatively EdU channel), followed by the "Analyze particle" function. Results of the latter analysis were exported as ROIs and visualized in the microglial channel for counting. For the cortex, quantification was done for each individual cortical layer defined using the Hoechst staining across a width of 500 μm, and BAMs of the meninges and perivascular space were excluded based on their location and morphology (Appendix Fig. S1). From postnatal stages, quantification was performed on all the cortical layers (I to VI). For the hippocampus, quantification was carried out in the CA1 and CA3 regions. While most slices were acquired at 10X using a 2 μm z-step, EDWM images were acquired on a 20μm-thick region at the center of the slice, with a 0.5 μm z-step, because of high microglial density in the EDWM. Quantifications were carried out on one or two images per brain. For comparison between controls and mutants, usually no blinding was done, since the phenotypes were visible.

For RNAscope, images were acquired with a microscope ECHO Revolve. For the quantification of *Csf1* expression after RNAscope experiments, z-stacks (with 2 μm intervals) were obtained from the CSA at E14.5 and from the EDWM at P7 using a Yokogawa CSUX1-A1 confocal spinning disk microscope with a 60x Nikon immersion objective and a Coolsnap HQ2 camera. These z-stacks

were processed in QuPath software (Bankhead et al, 2017). In QuPath, automatic annotation tools were used to create a full-image ROI. The "cell detection" function was applied to the 405 nm Hoechst channel to identify nuclei and estimate cell boundaries. The segmentation results were manually verified to ensure accurate cell detection. Microglia were identified based on IBA1 immunostaining performed on the same RNAscope-processed sections. The "subcellular detection" tool was then used to identify *Csf1* mRNA spots specifically within IBA1-positive cells. Parameters for detection were carefully adjusted and results inspected to ensure all *Csf1* mRNA probe dots were correctly identified. For microglia appearing across multiple optical sections, only the image with the maximum number of *Csf1* mRNA dots was used for quantification. The number of *Csf1* dots per IBA1-positive cell was retrieved using the "show detection measurements", specifically the "number of spots estimated" parameter.

### Statistical analysis

All data were expressed as mean ± SEM. According to the data structure, nonparametric tests were performed, namely two-sided paired or unpaired Mann–Whitney tests, two-sided Wilcoxon matched-pairs signed-rank tests, Kruskal–Wallis with Dunn's post hoc test or two-way ANOVA with Sidak's post hoc test, depending on whether single or multiple group comparisons with unpaired or paired controls were performed to assess differences. Chi-square test was used for comparison of microglia distribution across clusters and conditions. We have included both males and females throughout our experiments. To limit the number of animals we have not systematically compared males and females, albeit for critical experiments, such as assessing the density throughout development or the reduction in microglial density in *Csf1* ckO, we have compared males and females and did not observe statistically different distribution or phenotype (microglial density across females and males assessed through two-way ANOVA: cortex: $P = 0.7916$; striatum: $P = 0.8757$; POA: $P = 0.5840$; hippocampus: $P = 0.3631$ and microglial density in *Csf1* conditional $Cx3cr1^{creER}$ mutants across females and males assessed through unpaired Mann–Whitney tests $P = 0.70$). The numbers of females and males are indicated in the Source data for main figures and in the legends of the supplementary data for most of the experiments. All graphs and statistical analyses were generated using GraphPad Prism software (GraphPad Software Inc., USA), unless otherwise stated. ns not significant, $*P < 0.05$, $**P < 0.01$, $***P < 0.001$. Chi-square test was used for comparison of microglial distribution across clusters and conditions.

## Data availability

Raw and processed scRNA-seq data have been deposited in the ArrayExpress database (https://www.ebi.ac.uk/biostudies/ArrayExpress/studies/E-MTAB-14797) under accession number E-MTAB-14797. Microscopy data reported in this study will be shared by the lead contact upon request. Requests for further information and resources should be directed to and will be fulfilled by the lead contact, Morgane Sonia Thion (morgane.thion@college-de-france.fr).

The source data of this paper are collected in the following database record: biostudies:S-SCDT-10_1038-S44318-025-00625-8.

## Peer review information

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

## Acknowledgements

We thank Eléonore Touzalin, Amandine Delecourt and Carmen Le Moal, for assistance with mouse colonies, Marc Bajenoff for providing the *Csf1*^fl/fl mouse line, Claire Lansonneur for visualization of *Csf1R* and *Csf1* expression from the Hammond scRNA dataset (Hammond et al, 2019). We are grateful to members of the Thion and Garel teams for helpful discussions and critical review of the manuscript. We thank Lucy Robinson of Insight Editing London for scientific editing of the manuscript. We thank the IBENS Imaging Facility (France BioImaging, supported by ANR-10-INBS-04, ANR-10-LABX-54 MEMO LIFE, and ANR-11-IDEX-000-02 PSL* Research University, ("Investments for the future") and the Orion technological core (IMACHEM-IBiSA) of Center for Interdisciplinary Research in Biology for their support as well as the GDR Microglia and Neuroinflammation. This work was supported by grants to SG

and MST from INSERM, CNRS, ANR (ANR-19-CE16-0018 (SG), ANR-23-CE16-0033 (SG), ANR-23-CE16-0001 (MST), ANR-23-TERC-0017 (MST)), Fondation du Collège de France, Fondation pour la Recherche Médicale (FRM) (EQU202003010195, SG), Fédération pour la Recherche sur le Cerveau (FRC) (MST). CB is supported by an AMX PhD fellowship and the Fondation ARC pour la recherche sur le cancer (ARCDOC42022010004628); NO by the Fondation pour la Recherche Médicale (FRM, EQU202003010195). The development of the *Csf1*$^{fl/fl}$ mouse model was supported by grants to JXJ from the US National Institutes of Health grant AG045040 and the Welch Foundation grant AQ-1507. The development of the *Rosa*$^{Kir2.1}$ mouse model was supported by grants to GLB from ERC-2021-ADG-101054313.

## Author contributions

**Cécile Bridlance**: Conceptualization; Formal analysis; Investigation; Visualization; Methodology; Writing—original draft. **Sarah Viguier**: Formal analysis; Investigation; Visualization; Methodology. **Nicolas Olivié**: Formal analysis; Investigation; Visualization. **Edmond Dupont**: Formal analysis; Investigation; Visualization; Methodology. **Dorine Thobois**: Investigation. **Benjamin Mathieu**: Resources; Investigation. **Jean X Jiang**: Resources. **Guillermina López-Bendito**: Resources. **Melanie Greter**: Resources. **Burkhard Becher**: Resources. **Florent Ginhoux**: Resources. **Aymeric Silvin**: Resources; Investigation. **Esther Klingler**: Resources; Formal analysis; Visualization. **Sonia GAREL**: Conceptualization; Resources; Supervision; Funding acquisition; Visualization; Methodology; Writing—original draft; Project administration; Writing—review and editing. **Morgane Sonia Thion**: Conceptualization; Resources; Formal analysis; Supervision; Funding acquisition; Visualization; Methodology; Writing—original draft; Project administration; Writing—review and editing.

Source data underlying figure panels in this paper may have individual authorship assigned. Where available, figure panel/source data authorship is listed in the following database record: biostudies:S-SCDT-10_1038-S44318-025-00625-8.

## Disclosure and competing interests statement

The authors declare no competing interests.

# Expanded View Figures

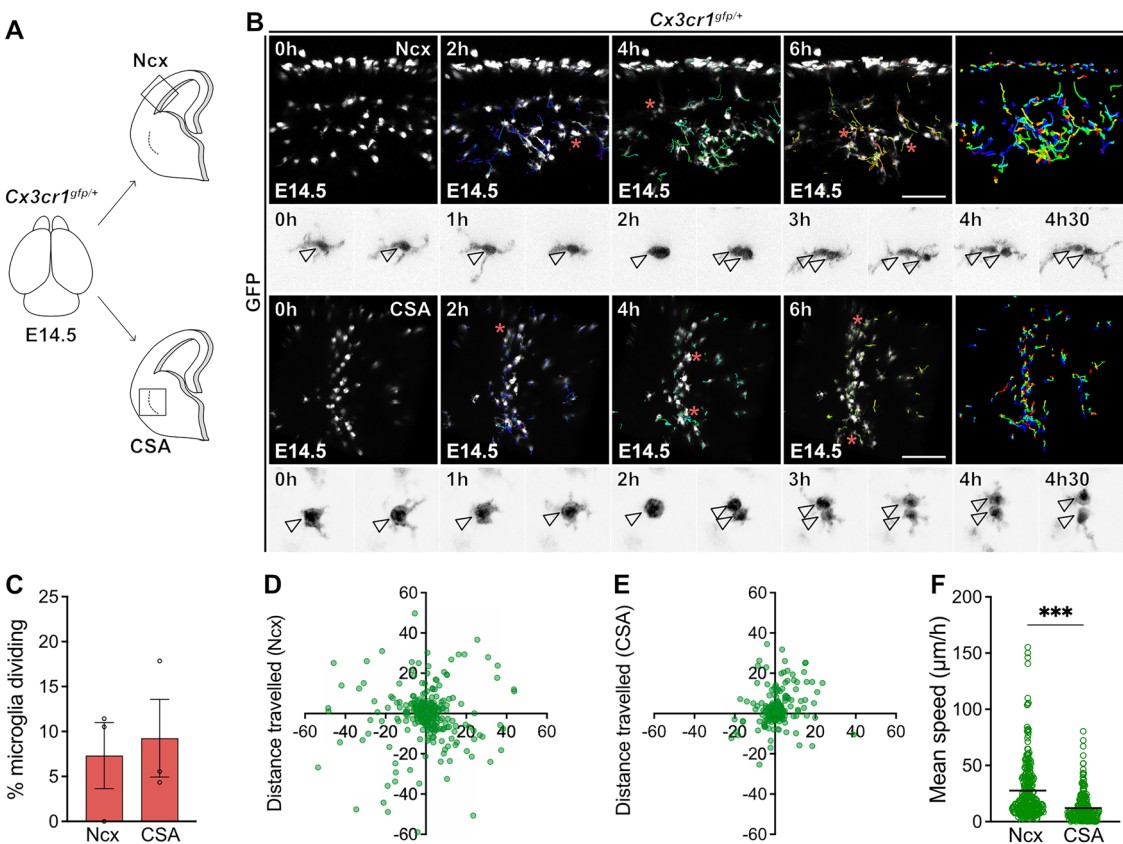

**Figure EV1.  Different dynamics of microglia in the embryonic neocortex and at the CSA.**

(A) Schematic representation of a *Cx3cr1^{gfp/+}* E14.5 coronal brain hemisection illustrating the areas where two-photon time-lapse imaging was performed. (B) Sample images with superimposed trajectories of individual cells from video tracking of microglia (GFP-positive cells) in E14.5 *Cx3cr1^{gfp/+}* embryos in the neocortex (top) or at the CSA (bottom) (length of recording, 6 h). Colors illustrate the speed of migration and stars highlight dividing microglia. Asterisks indicate single microglia dividing over the course of imaging. High magnification images track single microglia dividing over the course of imaging (open arrowheads). (C) Percentage of microglia dividing during the time-course of the recordings. (D, E) Distance traveled by individual cells tracked during the time-course of the recordings in the neocortex (D) or at the CSA (E). (F) Mean speed of individual cells tracked during the time-course of the recordings in the neocortex or at the CSA ($P < 0.0001$). Three embryos were imaged in each area from six distinct litters. Data were presented as mean ± SEM. Two-sided unpaired Mann–Whitney test (C) or Student's *t*-test (F) were performed to assess differences ns not significant, ***$P < 0.001$. Scale bars: 100 μm. CSA cortico-striato-amygdalar boundary, Ncx neocortex.

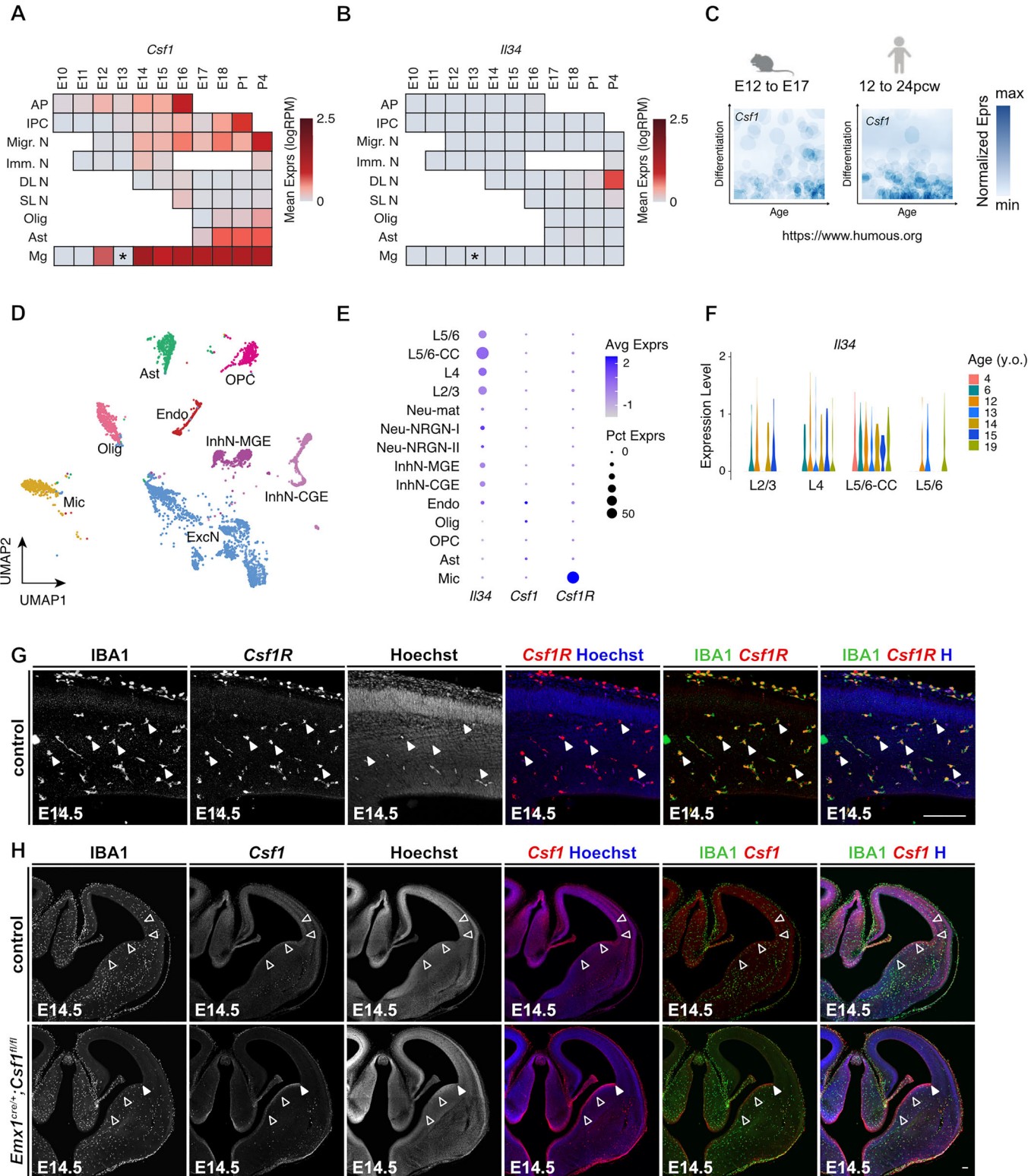

◀ **Figure EV2.  Spatiotemporal dynamics of *Csf1*, *Il34*, and *Csf1R* expression in the developing mouse and human cortex.**

(A, B) Heatmaps of single-cell gene expression of *Csf1* (A) or *Il34* (B) in neural cells of the neocortex and in microglia during development (mean reads per million). Data extracted from the available dataset produced by (Di Bella et al, 2021). Squares were left empty when data were not available. At E13 (marked by stars), only five microglial cells were identified. (C) Representative cyto-temporal gene expression landscape showing prenatal mouse and human *Csf1* expression in radial glia from HuMous.org (Javed et al, 2025). (D) Postnatal cortical human single-nucleus RNA-sequencing data represented in the UMAP space from (Baldassari et al, 2025). Original data were from (Velmeshev et al, 2023)(control individuals). (E) Dot plot showing the relative expression levels of *Il34*, *Csf1*, and *Csf1R* in postnatal cortical human cells. The color represents the normalized expression level across all cells within a cluster, while the dot size indicates the percentage of cells expressing each gene in that cluster. (F) Postnatal expression of *Il34* in human excitatory neuron types from brains of individuals aged 4 to 19 years. (G) RNAscope experiments on coronal brain sections from E14.5 control embryos showing *Csf1r* expression in IBA1-positive microglia in the neocortex (open arrowheads) ($n = 4$ from two litters). (H) RNAscope experiments on coronal brain sections from E14.5 control and *Emx1^cre;Csf1^{fl/fl}* embryos showing loss of *Csf1* expression (open arrowheads) in the neocortex of mutant mice (solid arrowhead) ($n_{control} = 5$; $n_{mutant} = 7$ from at least 3 litters). Scale bars: 100 μm (G, H). AP apical progenitors, Ast astrocytes, CGE caudal ganglionic eminence, DL N deep layer neurons, Imm. N immature neuron, IPC intermediate progenitor cell, MGE medial ganglionic eminence, Migr. N migrating neuron, Neu neuron, NRGN GluN-enriched protein neurogranin, Olig oligodendrocytes, pcw post-conceptional weeks, RPM reads per million, SL N superficial layer neurons.

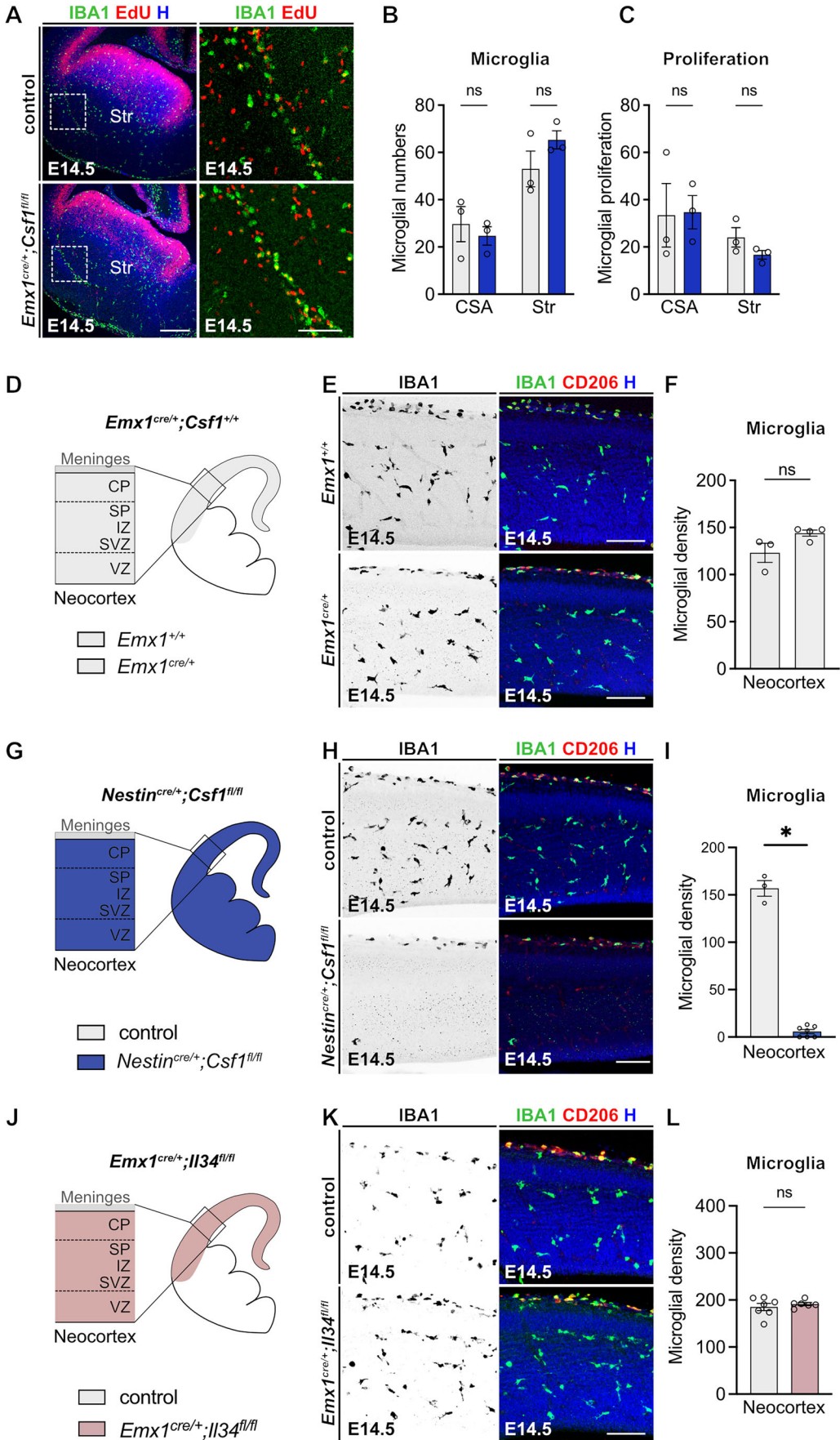

**Figure EV3. Impacts of *Csf1* and *Il34* on microglial density during early development.**

(A) Immunolabeling of coronal brain sections from E14.5 control and *Emx1*^cre/+^;*Csf1*^fl/fl^ embryos showing IBA1 and EdU used to identify proliferating microglia at the CSA and in the adjacent striatum. (B) Microglial numbers at the CSA and striatum of E14.5 control and *Emx1*^cre/+^;*Csf1*^fl/fl^ mice ($n_{control}$ = 3 M; $n_{mutant}$ = 3 F; from two distinct litters). (C) Microglial proliferation (EdU-positive) in the CSA and striatum of E14.5 control and *Emx1*^cre/+^;*Csf1*^fl/fl^ mice ($n_{control}$ = 3; $n_{mutant}$ = 3; from two distinct litters). (D) Schematic representation of a hemicoronal section of an E14.5 mouse telencephalon showing the pattern of recombination driven by the *Emx1*^cre^ line in the embryonic brain, including progenitors and excitatory neurons of the neocortex. (E) Immunolabeling of coronal brain sections from E14.5 control and *Emx1*^cre^ embryos showing IBA1 and CD206-positive cells in the neocortex. (F) Microglial densities in the neocortex at E14.5 ($n_{control}$ = 3 (2 F;1 M), $n_{cre}$ = 4 (1 F;3 M) from two distinct litters). (G) Schematic representation of a hemicoronal section of a E14.5 mouse telencephalon showing the pattern of recombination driven by the *Nestin*^cre^ line in the embryonic brain, including progenitors and excitatory neurons of the neocortex. (H) Immunolabeling of coronal brain sections from E14.5 control and *Nestin*^cre^;*Csf1*^fl/fl^ embryos showing IBA1 and CD206-positive cells in the neocortex. (I) Microglial densities in the neocortex at E14.5 ($n_{control}$ = 3 (2 M;1 F), $n_{homo}$ = 7 (4 M;3 F) from at least two distinct litters) ($P$ = 0.0167). (J) Schematic representation of a hemicoronal section of an E14.5 mouse telencephalon showing the pattern of recombination driven by the *Emx1*^cre^ line in the embryonic brain, including progenitors and excitatory neurons of the neocortex. (K) Immunolabeling of coronal brain sections from E14.5 control and *Emx1*^cre^;*Il34*^fl/fl^ embryos showing IBA1 and CD206-positive cells in the neocortex. (L) Microglial densities in the neocortex at E14.5 ($n_{control}$ = 7 (4 F;3 M), $n_{homo}$ = 6 (6 F) from two distinct litters). Data were presented as mean ± SEM. Two-sided unpaired Mann–Whitney test were performed to assess differences (B, C, F, I, L). ns not significant, *$P$ < 0.05. Scale bars: 500 μm (A, low mag); 100 μm (A, high mag), 200 μm (E, H, K). BAMs border-associated macrophages, CSA cortico-striatal-amygdalar boundary, CP cortical plate, IZ intermediate zone, SP subplate, Str striatum, SVZ subventricular zone, VZ ventricular zone.

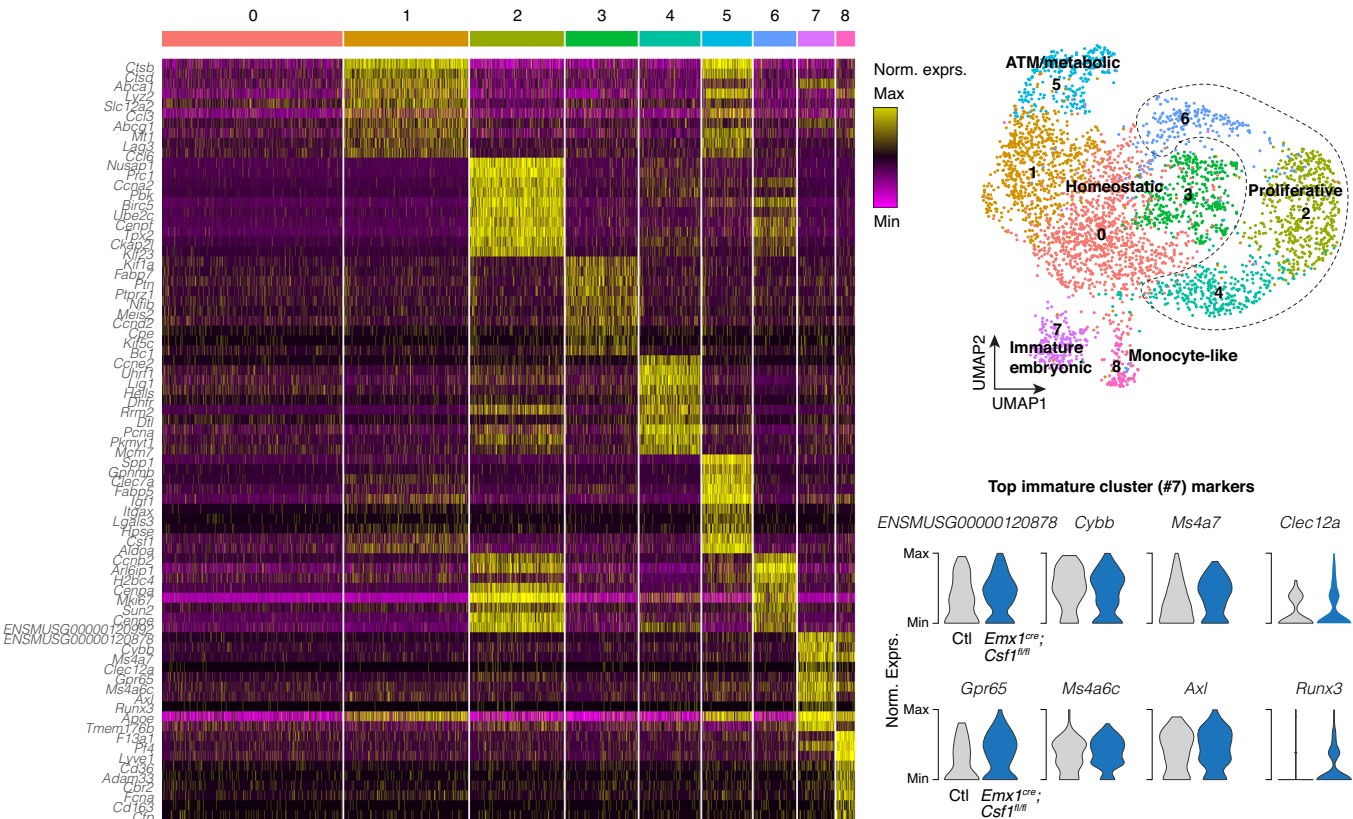

**Figure EV4. Annotation and marker expression of microglia cluster.**

Left, Expression of top ten markers (with avg_log2FC >1) for each microglial cluster across clusters. Right, Cluster annotation based on marker expression (top). Expression of top immature microglia markers in control and *Emx1^Cre^;Csf1^fl/fl^* cluster 7 cells (bottom). Source data are available online for this figure.

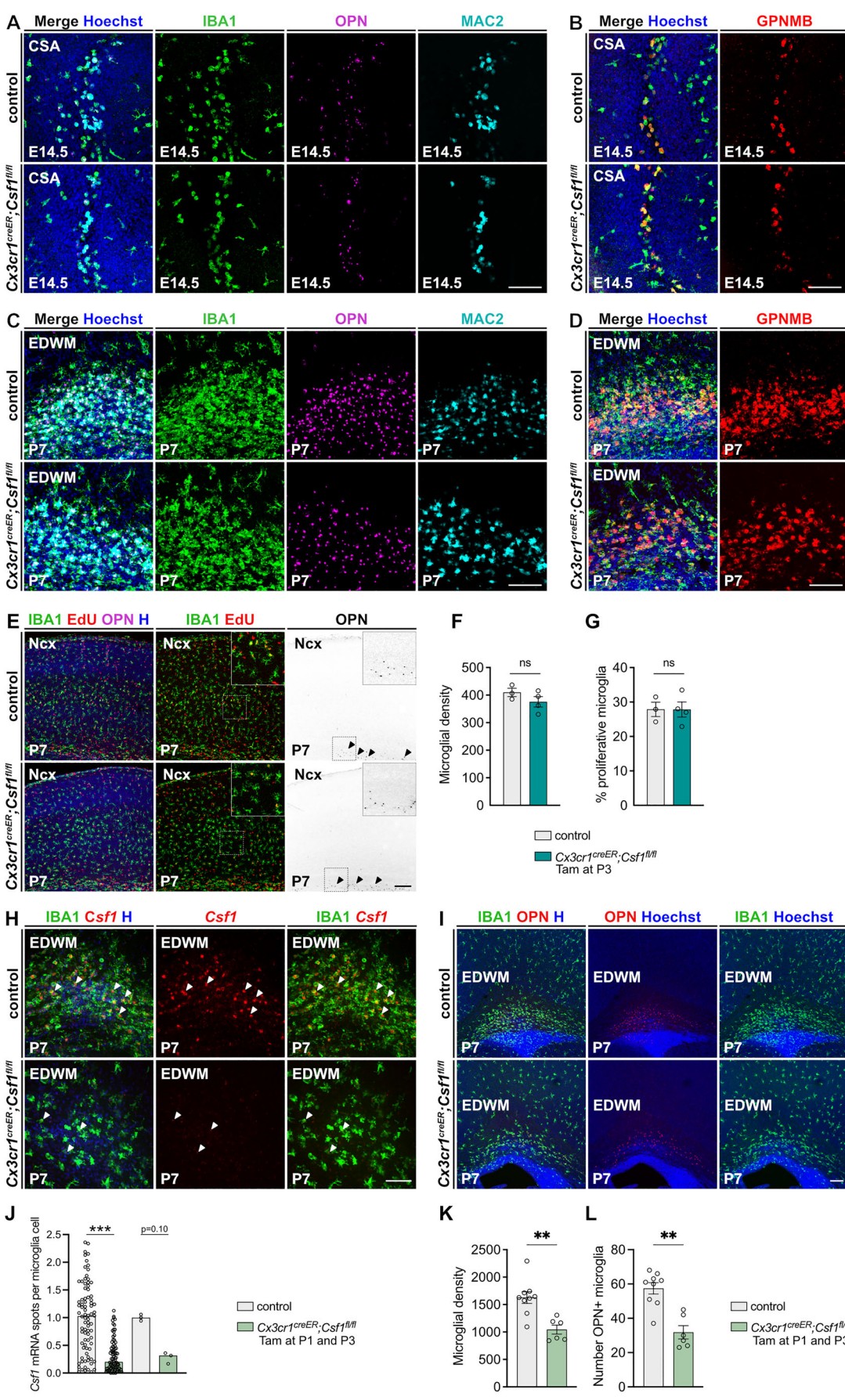

◀ **Figure EV5.  Microglial *Csf1* expression is specific to ATM accumulating in developmental hotspots and does not impact ATM core gene expression.**

(A) Immunolabeling of coronal brain sections showing comparable co-expression of GFP, MAC2, and OPN at the CSA in E14.5 control and *Cx3cr1^{creER/+};Csf1^{fl/fl}* embryos (tamoxifen administered at E12.5) ($n_{control} = 5$; $n_{mutant} = 4$). (B) Immunolabeling of coronal brain sections showing comparable co-expression of GFP and GPNMB at the CSA in E14.5 control and *Cx3cr1^{creER/+};Csf1^{fl/fl}* embryos (tamoxifen administered at E12.5). (C) Immunolabeling of coronal brain sections showing comparable co-expression of GFP, MAC2 and OPN at the CSA in P7 control and *Cx3cr1^{creER/+};Csf1^{fl/fl}* embryos (tamoxifen administered at P3) ($n_{control} = 3$; $n_{mutant} = 3$). (D) Immunolabeling of coronal brain sections showing comparable co-expression of GFP and GPNMB at the CSA in P7 control and *Cx3cr1^{creER/+};Csf1^{fl/fl}* embryos (tamoxifen administered at P3). (E) Immunolabeling of coronal brain sections from control and *Cx3cr1^{creER/+};Csf1^{fl/fl}* mice showing absence of expression of OPN in cortical microglia (IBA1-positive) at P7 (tamoxifen administered at P3) while some white matter microglia do express OPN (solid arrowheads). Close-ups are 200 µm wide. (F) Proportion of proliferative microglia in control and *Cx3cr1^{creER/+};Csf1^{fl/fl}* neocortex of P7 mice (tamoxifen administered at P3)($n_{control} = 3$ (2 F;1 M) ; $n_{mutant} = 4$ (2 F;2 M); from one litter). (G) Microglial density (IBA1-positive cells/mm²) in control and *Cx3cr1^{creER/+};Csf1^{fl/fl}* neocortex of P7 mice (tamoxifen administered at P3)($n_{control} = 3$ (2 F;1 M) ; $n_{mutant} = 4$ (2 F;2 M); from one litter). (H) RNAscope experiments on coronal brain sections from P7 control and *Cx3cr1^{creER/+};Csf1^{fl/fl}* pups showing *Csf1* expression in IBA1-positive ameboid microglia at the CSA (solid arrowheads) after tamoxifen administration at P1 and P3. (I) Normalized quantification of RNAscope experiments on coronal brain sections from P7 control and *Cx3cr1^{creER/+};Csf1^{fl/fl}* embryos showing decreased *Csf1* expression in mutant mice. Left column shows all quantified cells and right column the mean expression per animal ($n_{control} = 3$ (1 F, 2 M); $n_{mutant} = 3$ (2 M,1 F); from two distinct litters). (J) Immunolabeling of coronal brain sections from control and *Cx3cr1^{creER/+};Csf1^{fl/fl}* mice showing co-expression of IBA1 and OPN in ameboid microglia within the EDWM accumulation at P7 (tamoxifen administered at P1 and P3) ($n_{control} = 3$ (1 F,1 M) ; $n_{mutant} = 3$ (2 F;1 M); from one litter)(all cells, left: $P < 0.0001$; mean per animal, right: $P = 0.10$). (K) Microglial density (IBA1-positive cells/mm²) in control and *Cx3cr1^{creER/+};Csf1^{fl/fl}* EDWM of P7 mice (tamoxifen administered at P1 and P3) ($n_{control} = 9$ (3 F;6 M); $n_{mutant} = 6$ (3 F;3 M); from three distinct litters) ($P = 0.0028$). (L) Number of OPN-positive microglia at the EDWM at P7 ($n_{control} = 9$ (3 F;6 M); $n_{mutant} = 6$ (3 F;3 M); from three distinct litters) ($P = 0.0016$). Data were presented as mean ± SEM. Two-sided unpaired Mann–Whitney test were performed to assess differences (F, G, J–L). ns not significant, **$P < 0.01$, ***$P < 0.001$. Scale bars: 100 µm (A–D, H, I); 200 µm (E). CSA cortico-striatal-amygdalar boundary, EDWM early-dorsal white matter, H Hoechst, Ncx neocortex.

