## [Peer Review File · The EMBO Journal]

Microglial colonization of the developing mouse brain is controlled by both microglial and neural CSF1

Cécile Bridlance, Sarah Viguier, Nicolas Olivié, Edmond Dupont, Dorine Thobois, Benjamin Mathieu, Jean Jiang, Guillermina Lopez-Bendito, Melanie Greter, Burkhard Becher, Florent Ginhoux, Aymeric Silvin, Esther Klingler, Sonia GAREL, and Morgane Thion

Corresponding author(s): Morgane Thion (morgane.thion@college-de-france.fr) , Sonia GAREL (garel@biologie.ens.fr)

Review Timeline:

Submission Date:	22nd Jan 25
Editorial Decision:	5th Mar 25
Revision Received:	25th Jul 25
Editorial Decision:	21st Aug 25
Revision Received:	23rd Sep 25
Accepted:	28th Sep 25

Editor: Ioannis Papaioannou

Transaction Report:

Dear Dr. Thion,

Thank you for submitting your manuscript EMBOJ-2025-120269 for consideration by The EMBO Journal, and for your patience during peer review. Your manuscript has now been seen by three experts in the field, and we have received the full set of their comments, which you can find below.

As you will see, all three referees recognize that the work and the manuscript are well-performed and written, reporting intriguing and compelling findings, and providing important new insights into the role of neural-derived CSF1 in controlling microglial colonization in the developing neocortex that would be of broad interest to those working in the fields of neurodevelopment and neuroimmunology. They also identify a number of limitations, however, and they point out a number of aspects that require additional experimental validation, clarification, and discussion. They all provide detailed suggestions for the improvement of the work and the manuscript, which we think would strengthen the work further and increase its impact on the field.

Given the referees' positive comments and recommendations, I would like to invite you to submit a revised version of the manuscript along with a detailed point-by-point response addressing all referees' comments. I should add that it is The EMBO Journal policy to allow only a single round of major revision, and acceptance of your manuscript will therefore depend on the completeness of your responses in this revised version. Please let me know if you have any questions or comments that you would like to discuss with me.

We generally allow three months as standard revision time (June 4, 2025). As a matter of policy, competing manuscripts published during this period will not negatively impact our assessment of the conceptual advance presented by your study. However, we request that you contact us as soon as possible upon publication of any related work, to discuss how to proceed. Should you foresee a problem in meeting this three-month deadline, please let us know in advance and we may be able to grant an extension.

Thank you for the opportunity to consider your work for publication in The EMBO Journal. I look forward to your revision.

Best regards,

Ioannis

Instructions for preparing your revised manuscript

1. When you are ready to submit the revision, please upload:

- A Word file of the manuscript text (including legends of main Figures, EV Figures and Tables). Please make sure that changes are highlighted (or "tracked") to be clearly visible.

- Individual production-quality figure files (one file per figure). When assembling your figures, please refer to our figure preparation guidelines in order to ensure proper formatting and readability in print as well as on screen:

If the data shown in a figure are obtained from n {less than or equal to} 2, please use scatter plots showing the individual data points.

i. the name of the statistical test used to generate error bars and P values

ii. the number (n) of independent experiments (please specify technical or biological replicates) underlying each data point

(discussion of statistical methodology can be reported in the Materials and Methods section, but figure legends should contain a basic description of n , P , and the test applied)

iii. the nature of the bars and error bars (s.d., s.e.m.).

- A point-by-point response to the referees' comments, with a detailed description of the changes made (as a word file). All

referees' concerns must be fully addressed and their suggestions taken on board. When preparing your letter of response to the referees' comments, please bear in mind that this will form part of the Review Process File and will therefore be available online to the community. Please note that you have the possibility to opt out of the transparent process at any stage prior to publication by letting the editorial office know (contact@embopress.org); if you do opt out, the Review Process File link will point to the following statement: "No Review Process File is available with this article, as the authors have chosen not to make the review process public in this case.". For more details on our Transparent Editorial Process, please visit our website: <https://www.embopress.org/page/journal/14602075/authorguide#transparentprocess>

- Expanded View (EV) files (replacing Supplementary Information) that are collapsible/expandable online. A maximum of 5 EV Figures can be typeset. EV Figures should be cited as "Figure EV1, Figure EV2" etc. in the text, and their respective legends should be included in the manuscript file after the legends of regular figures. See detailed instructions regarding Expanded View files here:

- For the figures that you do NOT wish to display as Expanded View figures, they should be bundled together with their legends in a single PDF file called "Appendix", which should start with a short Table of Contents (including page numbers). Appendix figures should be referred to in the main text as: "Appendix Figure S1, Appendix Figure S2" etc. Please see detailed instructions here: <https://www.embopress.org/page/journal/14602075/authorguide#expandedview>

- A complete author checklist, which you can download from our author guidelines (<https://www.embopress.org/page/journal/14602075/authorguide>). Please note that the checklist will also be part of the Review Process File.

2. Please note that no statistics should be calculated and shown in Figures if $n=2$. Please also note that each p value should be reported as an exact value.

3. Before submitting your revision, primary datasets (and computer code, where appropriate) produced in this study need to be deposited in appropriate public databases (see <https://www.embopress.org/page/journal/14602075/authorguide#dataavailability>). In particular, we kindly request you to deposit all RNA-seq data produced in the study. The accession numbers, database, and the specific URLs (links) should be listed in a formal "Data availability" section (placed after Methods), following the example below:

"The RNA-seq datasets produced in this study are available in the following database:
Gene Expression Omnibus GSE46843 (<https://www.ncbi.nlm.nih.gov/geo/query/acc.cgi?acc=GSE46843>)"

*** All links should resolve to a page where the data can be accessed. ***

*** Please remember to provide in the Data availability section of your revised manuscript reviewer passwords if the datasets are not yet public. ***

*** The Data Availability Section is restricted to new primary data that are part of this study. In case you have no data that require deposition in a public database, please state so instead of referring to the database: "Our study includes no data deposited in public repositories." under the heading "Data availability". ***

4. The materials and methods need to be described in the manuscript using our structured methods format, which is now required for all research articles. According to this format, the Methods section includes a single "Reagents and Tools Table" - listing key reagents, experimental models, software and relevant equipment including their sources and relevant identifiers- followed by a "Methods and Protocols" section describing the methods. Please download and fill our Reagents and Tools Table template (.docx), which you can find in our author guide:

<https://www.embopress.org/page/journal/14602075/authorguide#structuredmethods>. When submitting your revised manuscript, please do not include the Reagents and Tools Table in the Methods section of the manuscript but instead upload it as a separate file choosing the file type "Reagent Table".

5. Please check that the title and the abstract of the manuscript are brief, yet explicit, even to non-specialists. The length of the title should not exceed 100 characters, and the abstract should be a single paragraph not exceeding 175 words.

6. Please also note our reference format: <https://www.embopress.org/page/journal/14602075/authorguide#referencesformat>.

8. Please remember: digital image enhancement is acceptable practice, as long as it accurately represents the original data and conforms to community standards. If a figure has been subjected to significant electronic manipulation, this must be noted in the

figure legend or in the "Materials and Methods" section. The editors reserve the right to request original versions of figures and the original images that were used to assemble the figure.

9. Our journal encourages inclusion of data citations in the reference list to directly cite datasets that were obtained from public databases. Data citations in the article text are distinct from normal bibliographical citations and should directly link to the database records from which the data can be accessed. In the main text, data citations are formatted as follows: "Data ref: Smith et al, 2001" or "Data ref: NCBI Sequence Read Archive PRJNA342805, 2017". In the Reference list, data citations must be labeled with "[DATASET]". A data reference must provide the database name, accession number/identifiers, and a resolvable link to the landing page from which the data can be accessed at the end of the reference. Further instructions are available at: <https://www.embopress.org/page/journal/14602075/authorguide#referencesformat>.

10. We request authors to consider both actual and perceived competing interests. Please review our policy (<https://www.embopress.org/page/journal/14602075/authorguide#conflictsofinterest>) and update your competing interests statement if necessary. Please name this section 'Disclosure and competing interests statement' and place it after the Acknowledgements section.

11. Please note that all corresponding authors are required to provide an ORCID ID upon submission of a revised manuscript (<https://orcid.org/>). Please find instructions on how to link your ORCID ID to your account in our manuscript tracking system in our Author guidelines (<https://www.embopress.org/page/journal/14602075/authorguide#authorshipguidelines>).

12. We use CRediT to specify the contributions of each author in the journal submission system. CRediT replaces the author contribution section, which should be removed from the manuscript. Please use the free text box to provide more detailed descriptions. See also guide to authors: <https://www.embopress.org/page/journal/14602075/authorguide#authorshipguidelines>.

14. We would also welcome the submission of cover suggestions or motifs to be used by our Graphics Illustrator in designing a cover.

15. Please use the link below to submit your revision:
<https://emboj.msubmit.net/cgi-bin/main.plex>

Referee #1:

Summary and General Assessment:

The manuscript "Microglial colonization is shaped by intrinsic and extrinsic CSF-1 during early brain development" presents a comprehensive study on the developmental colonization of microglia in the mouse brain, emphasizing the role of CSF-1 derived from neural and microglial sources. The authors use a combination of genetic tools, immunofluorescence, live imaging, and single-cell RNA sequencing to map microglial proliferation dynamics and spatial distribution. The key finding is that microglia depend on local sources of CSF-1 from neuronal populations for their proliferation and maintenance, with ATM-like microglia exhibiting intrinsic CSF-1 expression to sustain their proliferation. This study provides novel insights into the mechanisms underlying microglial colonization, with potential implications for neurodevelopmental and neurodegenerative disorders. While the study is well-executed and provides valuable contributions to the field, several aspects require clarification and additional validation. Below are specific points categorized into major and minor concerns.

Major Comments:

Novelty and Significance:

The authors claim that microglia rely on distinct sources of CSF-1 for their proliferation and distribution. While this is an important finding, the novelty should be better articulated in the context of previous studies on CSF-1R signaling.

The relevance of these findings to human brain development should be discussed further.

While the local, transient action of neural Csf1 is convincingly demonstrated, the functional consequences of this mechanism in the adult brain remain speculative. Further discussion on the long-term implications of disrupted microglial proliferation patterns would enhance the impact of the study.

Experimental Design and Validation:

The Cx3cr1creER; Csf1fl/fl model used to assess microglial-intrinsic Csf1 expression shows only partial recombination as the authors state also in their limitation section. The authors should perform additional validation (e.g., qPCR or RNAscope quantification) to confirm the extent of Csf1 loss.

The study relies heavily on the *Emx1cre; Csf1fl/fl* model to demonstrate the role of neural CSF-1. However, this model affects multiple neural populations. I am wondering, whether the authors considered using more cell-type-specific Cre lines (e.g., Nestin-Cre, Neurog2-Cre), and if so what was the rationale sticking to the *Emx1cre; Csf1fl/fl* model.

Mechanistic Insights:

The finding that ATM-like microglia sustain their proliferation via intrinsic CSF-1 expression is intriguing. However, the functional consequences of this self-sustaining proliferation remain unclear. Does this mechanism contribute to long-term microglial density regulation or have implications for disease states?

The role of IL-34 is somewhat underexplored, despite its known importance in microglial maintenance. The authors should clarify whether it contributes to microglial repopulation in the *Emx1cre; Csf1fl/fl* model.

Clarity of Data Interpretation:

In Figure 4, microglial repopulation in the *Emx1cre; Csf1fl/fl* model is discussed as evidence of transient CSF-1 dependence. However, the underlying mechanisms of this repopulation (proliferation vs. migration) are not well defined. Additional experiments, such as fate-mapping of remaining microglia, would strengthen the conclusions.

Minor Comments:

Writing and Organization:

The manuscript is generally well-written but could benefit from streamlining in the results section to improve clarity.

The discussion occasionally reiterates points already made in the results. Condensing this section while maintaining key insights would improve readability.

Some claims, such as the generalization of findings to human microglia, should be softened or supported by additional evidence. The discussion should clearly delineate which findings are specific to mice and which might be extrapolated.

The manuscript states that both male and female mice were used in the experiments. However, the reporting of sex differences is inconsistent. While the text explicitly mentions that no differences in microglial proliferation or density were observed between males and females in some experiments, it is unclear whether all experiments were conducted with a balanced representation of both sexes or whether some used only a mixed cohort without sex-specific analysis.

To improve clarity, the authors should:

Clearly specify the sex of animals used for each experiment in the Methods section and figure legends.

Indicate whether data from males and females were pooled or analyzed separately. If pooled, justify why no sex-based differences were considered.

Explicitly state when statistical analyses for sex differences were performed, and if they were not conducted, provide a rationale.

Figures and Data Presentation:

Some figure panels (e.g., Figure 1G-J) are dense and difficult to interpret. The use of additional annotations or simplified schematics would help readers grasp key findings more easily.

Figure S7 should be integrated into the main figures to highlight the ATM-specific expression of CSF-1 more prominently.

Reproducibility and Methods:

The methods section is thorough, but some aspects (e.g., quantification strategy for RNAscope) should be more explicitly detailed to ensure reproducibility.

The authors mention statistical analyses but should provide effect sizes for key comparisons to better contextualize their findings.

Conclusion:

This study provides significant insights into the developmental dynamics of microglial colonization and proliferation, particularly the role of CSF-1 from distinct cellular sources. The findings are compelling and will be of broad interest to neurodevelopmental and neuroimmunology researchers. However, additional experimental validation and improved clarity in data interpretation are needed to fully substantiate the claims. Addressing these points will strengthen the manuscript and enhance its impact.

Referee #2:

In this manuscript, Bridance et al. aim to show how microglial colonization of the developing mouse brain occurs. The authors provide data that suggest microglial proliferation requires both extrinsic growth factor from neurons and intrinsic growth factor from axon tract-associated microglia (ATMs), but does not require normal neuronal activity. Given the importance of microglia as innate immune cells of the brain that regulate neural networks and are central to brain health and disease, understanding how microglia develop properly is important and of widespread interest. In this report, the authors use *Cx3cr1gfp/+* mice and perform brain immunohistochemistry with proliferative markers at successive developmental time points with beautiful microscopy images to characterize microglial proliferation over time in the developing brain. Then, using Cre-lox genetic recombination system in *Emx1Cre/+ ; Kir2.1* (floxed stop), *Emx1Cre/+ ; Csf1fl/fl*, and *Cx3cr1Cre/+ ; Csf1fl/fl* mice the authors interrogate the roles of neural activity, neural *Csf1*, and microglial *Csf1* on microglial colonization. Lastly, the authors perform scRNAseq and RNA in situ hybridization to further characterize how developing microglial transcriptomes change without *Csf1* from *Emx1*-expressing cells. While these studies are well-conducted and of potentially broad interest to the neuroscience community, there are two major concerns that if addressed would greatly improve the impact of this work

1) Neuronal activity is not required for microglial colonization. Using *Emx1Cre/+ ; Kir2.1* (floxed stop) mice, the authors find no change in microglial colonization in response to presumed hyperpolarization of the circuit. They follow with whisker plucking or infraorbital nerve injury to show no changes in colonization. Given that these sensory deprivation experiments can lead to increased pruning and thus provide a theoretical counterbalance to any potential depletion, it becomes important that the

authors prove hyperpolarization in their circuit with the Emx1Cre⁺ model by citation or physiology. In addition, while density was unchanged there is certainly a different characteristic noted in the Figure 2G particularly throughout which calls into question the observation based on density. Perhaps absolute number count or some higher power imaging can help adjudicate the Emx1 Cre Kir experiments and strengthen the argument.

2) Neuronal Csf1. The authors conclude that Csf1 expression "later in astrocytes and oligodendrocytes" was targeted using their Emx1Cre⁺; Csf1^{fl/fl} mice; there are no validation experiments demonstrating specificity of Csf1 reduction. Because the authors conclude neural Csf1 is required for microglial proliferation, the authors need to show validation of their knockout. Specifically, the authors could perform RNA in situ for Csf1 transcripts along with astrocyte, neuron, and oligodendrocyte markers to quantify the impact of Csf1 depletion of microglial colonization or clarify that it is parenchymal extrinsic Csf1 rather than neuronal.

Minor concerns.

1) Morphological changes. In the text of the manuscript, the authors refer to amoeboid and ramified morphologies in developmental hotspots. To strengthen these claims, the authors should include quantification of these morphologies: number or length of processes, individual fractal or skeletal analysis, or cell body area/perimeter analysis. For example: PMID: 36307475

2) In panels 1L and 1N, the graphs do not extend to the P14 timepoint even though the text body suggests that at P14 proliferation becomes undetectable. These data should be included.

3) Lastly, the conclusions of this paper suggest that there is a dose-dependent effect but also a cell-type specific effect of CSF1 production - there is no evidence to suggest the dose dependent effect. The authors could quantify Csf1 expression in their models as suggested above or overexpress CSF1 in a non-neuronal cell type or even simply inject more CSF1 in the Emx1Cre⁺; Csf1^{fl/fl} mice.

4) Cre controls - in knock-in knockout systems, assessing the impact of Cre is important in models. Suggest the authors in the future include Cre only controls in their analyses to strengthen their conclusions. PMID: PMC5910175

Referee #3:

Bridance et al. demonstrate that microglia undergo two distinct waves of proliferation during forebrain development in mice, gathering transiently in specific hotspots where they adopt a proliferative Axon-Tract-associated Microglia (ATM)-like phenotype. The authors convincingly show that prenatal and early postnatal colonization patterns of microglia are largely independent of neuronal activity. Using several transgenic mouse models with conditional CSF1 inactivation, the authors provide clear evidence that embryonic cortical microglia critically depend on locally produced neural-derived CSF1, predominantly originating from cortical progenitor cells and, to a lesser extent, post-mitotic neurons. This action of CSF1 is local, dose-dependent, and transient. Notably, intrinsic CSF1 expression by ATM itself contributes to sustaining their proliferation within developmental hotspots.

The paper is well-written, structured logically, and significantly clarifies the dynamics of microglial colonization in the developing brain. Furthermore, it opens new avenues for investigating the region- and stage-specific roles of microglia during embryonic brain development.

Novelty: While the reliance of microglia on CSF1 for proliferation and survival has been previously documented, including the role of neuronal-derived CSF1 in microglial number colonisation regulation in late embryonic and postnatal cerebellar development, this study uniquely illustrates the critical role of neural-derived CSF1 in controlling microglial colonization and proliferation specifically in the developing neocortex. Additionally, the authors confirm the role for ATM microglia in producing intrinsic CSF1, and demonstrate it further promotes their own proliferation.

Major Comments:

1. It would be valuable for the authors to clarify if microglial colonization in the neocortex depends on a CSF1 gradient. Does the local CSF1 gradient influence microglial migration in addition to proliferation?

2. Given prior evidence of CSF1R expression on neuronal progenitors and cortical neurons, it is important to address if the presented CSF1 effect is direct or indirect. Does neuronal-CSF1 stimulate neuronal populations that might in turn influence microglial dynamics?

3. Could the authors assess if there are other (neuronal or glial) changes beyond microglial numbers in the Emx1cre⁺;Csf1fl/+ and Emx1cre⁺;Csf1fl/fl models? Performing transcriptomic analyses on neuronal and glial populations or bulk would greatly enhance understanding and help address potential indirect effects.

Response to Reviews on Manuscript EMBOJ-2025-120269 by Bridlance et al.

We thank the reviewers and the Editor for their thoughtful comments and recognition of the interest and importance of our study. We have now strengthened the manuscript in line with the points raised during review and believe that we have conclusively answered the questions raised. To do so, we have generated and included additional experimental data that firmly support our original conclusions, shown in **Figures 4, 6, EV2, EV3 and EV5** alongside more in-depth data analyses and enhancements of the main text as detailed in the point-by-point response below.

Referee #1:*Summary and General Assessment:*

The manuscript "Microglial colonization is shaped by intrinsic and extrinsic CSF-1 during early brain development" presents a comprehensive study on the developmental colonization of microglia in the mouse brain, emphasizing the role of CSF-1 derived from neural and microglial sources. The authors use a combination of genetic tools, immunofluorescence, live imaging, and single-cell RNA sequencing to map microglial proliferation dynamics and spatial distribution. The key finding is that microglia depend on local sources of CSF-1 from neuronal populations for their proliferation and maintenance, with ATM-like microglia exhibiting intrinsic CSF-1 expression to sustain their proliferation. This study provides novel insights into the mechanisms underlying microglial colonization, with potential implications for neurodevelopmental and neurodegenerative disorders.

While the study is well-executed and provides valuable contributions to the field, several aspects require clarification and additional validation. Below are specific points categorized into major and minor concerns.

We are grateful to the reviewer for acknowledging the strength, interest and novelty of our findings and for providing such constructive feedback.

*Major Comments:**Novelty and Significance:*

The authors claim that microglia rely on distinct sources of CSF-1 for their proliferation and distribution. While this is an important finding, the novelty should be better articulated in the context of previous studies on CSF-1R signaling.

We thank the reviewer for identifying the need to more strongly highlight the novelty of our findings within the existing literature on CSF-1R signaling. Accordingly, we have revised the text in the Introduction and Discussion to more clearly illustrate how our data build upon and extend previous studies. Specifically, we now emphasize that while the CSF-1R signaling has long been recognized as essential for microglial survival and development, our work reveals a region- and stage-specific dependence on distinct cellular sources of *Csf1*, i.e., extrinsic neural cells within the neocortex versus intrinsic microglia within specific hotspots of proliferating microglia. These insights underscore the spatial, temporal and cell-type complexity of *Csf1*-mediated microglial regulation and represent a key advance in understanding how local CNS environments shape microglial distribution and proliferation during early brain development.

The relevance of these findings to human brain development should be discussed further.

To enable us to discuss the relevance of our findings to human brain development from an informed perspective, we have now analyzed two additional publicly available and complementary datasets: first, we used HuMous.org, an interactive platform that provides longitudinal single-cell RNA-seq data from the developing mouse and human neocortex¹; and, second, we examined a postnatal single-nucleus RNA-seq dataset from the human prefrontal (and partially parietal) cortex, covering individuals aged 4 to 19 years^{2,3}.

Consistent with our observations in the mouse, *Csf1* expression in humans was highest in prenatal radial glial progenitors, while *Il34* was not detected during prenatal stages. However, *Il34* is clearly expressed postnatally in excitatory neurons, in line with a role that emerges after birth. Importantly, *Csf1r* was not expressed in non-microglial cells during development, as confirmed both in scRNA-seq datasets and by RNAscope analysis in the E14.5 mouse neocortex.

These results highlight a conserved temporal and cell-type-specific expression pattern of *Csf1*, *Il34*, and *Csf1r* between mouse and human brains, reinforcing the relevance of our functional analyses in mice for understanding human brain development. This new analysis has been incorporated into the revised manuscript and is now presented in **Figure EV2A-F**, and included in the revised Discussion.

While the local, transient action of neural Csf1 is convincingly demonstrated, the functional consequences of this mechanism in the adult brain remain speculative. Further discussion on the long-term implications of disrupted microglial proliferation patterns would enhance the impact of the study.

We agree that a deeper discussion of the long-term implications of disrupted microglial proliferation patterns would strengthen the manuscript. Accordingly, we have revised the Discussion to better address this point. We now elaborate on how transient, localized disruptions of *Csf1* expression during critical developmental windows may have enduring consequences for microglial maturation, and so for long-term function. Although we acknowledge that our current study focuses primarily on developmental stages, we propose that these findings provide a framework for future investigations into how early microglial perturbations may have long-lasting consequences potentially contributing to brain disorders later in life.

Experimental Design and Validation:

The Cx3cr1creER; Csf1fl/fl model used to assess microglial-intrinsic Csf1 expression shows only partial recombination as the authors state also in their limitation section. The authors should perform additional validation (e.g., qPCR or RNAscope quantification) to confirm the extent of Csf1 loss.

Using RNAscope experiments, we have now quantified the decreased expression of *Csf1* in microglia at the CSA and EDWM of mutant mice compared to controls. This approach confirmed that, while we do have a significant reduction, the recombination is not fully efficient, especially at the embryonic CSA. These quantifications are now included in the main manuscript (**Figure 6C,E**).

The study relies heavily on the Emx1cre; Csf1fl/fl model to demonstrate the role of neural CSF-1. However, this model affects multiple neural populations. I am wondering, whether the

authors considered using more cell-type-specific Cre lines (e.g., Nestin-Cre, Neurog2-Cre), and if so what was the rational sticking to the Emx1cre;Csf1fl/fl model.

We thank the reviewer for this constructive comment. In the original submission, we also included data from the *Nes^{cre}; Csf1^{fl/fl}* model, which allowed us to inactivate *Csf1* only in post-mitotic neurons (Figure 3K-N). By comparing this model with the *Emx1^{cre}; Csf1^{fl/fl}* one, which recombines in neocortical progenitors and their progeny, we aimed to determine the specific contribution of *Csf1* expressed by neural progenitors versus post-mitotic neurons within the neocortex. This comparison allowed us to make two key observations: (1) that neuronal progenitors are the main source of *Csf1* during embryogenesis; and (2) that the absence of *Csf1* in postmitotic neurons is not sufficient to maintain normal microglial levels, indicating limited compensatory potential even in a small region like the neocortex.

Additionally, we performed further experiments using the *Nestin^{Cre}; Csf1^{fl/fl}* model, which targets a broad population of neural precursors (Figure EV3G-I). These experiments revealed a similarly drastic depletion of microglia at E14.5, comparable to that observed with the *Emx1^{cre}; Csf1^{fl/fl}* model, reinforcing our findings that local neural *Csf1* is essential for embryonic microglia, rather than circulating *Csf1*.

We thus focused on the *Emx1^{cre}; Csf1^{fl/fl}* model as it allowed a more spatially restricted depletion of *Csf1* within the neocortex as compared to in *Nestin^{Cre}; Csf1^{fl/fl}* mice (Figure EV2G,H). This targeted approach enabled us to investigate whether the absence of *Csf1* in a highly localized environment could be compensated for by adjacent brain regions, which importantly was not the case. Thus, the *Emx1^{cre}; Csf1^{fl/fl}* model offers the advantage of inducing a robust, drastic, and regionally confined microglial depletion, which we believe is particularly valuable for further dissecting the local and specific roles of microglia during development.

Mechanistic Insights:

The finding that ATM-like microglia sustain their proliferation via intrinsic CSF-1 expression is intriguing. However, the functional consequences of this self-sustaining proliferation remain unclear. Does this mechanism contribute to long-term microglial density regulation or have implications for disease states?

We agree with the reviewer that this is particularly intriguing. We have tried now through several crosses to stably inactivate *Csf1* in microglia or macrophages, notably using the *Fcrls^{cre}* line, or *Cx3cr1^{creER}* using high doses of hydroxytamoxifen. Unfortunately, we could not drastically increase the percentage of recombination induced, potentially due to the fact that the two loxP sites are located far away from each other in the *Csf1fl* construct⁴, making recombination likely more difficult to be achieved⁵. To nonetheless explore the consequences of *Csf1* intrinsic inactivation, we attempted to improve *Csf1* inactivation by performing two tamoxifen injections in conditional mutants generated with the *Cx3cr1^{creER}* line, at P1 and P3, and examine microglial hotspot in the EDWM (Figure EV5H-K). After confirming a stronger decrease in *Csf1* expression using RNAscope (Figure EV5H,J), we observed a clear reduction in microglial and ATM accumulations in the P7 EDWM of conditional mutants compared to controls (Figure EV5I,K). These new results significantly reinforced the notion that intrinsic *Csf1* plays a role in microglial hotspots. Finally, we additionally discuss how a cell-autonomous effect of *Csf1* expression could be of importance in the context of Disease Associated Microglia, with implications for disease progression.

The role of IL-34 is somewhat underexplored, despite its known importance in microglial maintenance. The authors should clarify whether it contributes to microglial repopulation in the Emx1cre; Csf1fl/fl model.

We thank the reviewer for their constructive suggestion and have now investigated *Il34* contribution to microglial repopulation using a complementary genetic model. After several rounds of backcrossing, we generated and analyzed *Emx1^{cre}; Csf1^{fl/fl}; Il34^{fl/fl}* double conditional knockout mice (**Figure 4C-E**). In this model, we observed a drastic reduction in microglial density at P7, highlighting the critical role of *Il34* in microglial repopulation in *Emx1^{cre}; Csf1^{fl/fl}* animals.

Consistent with previous findings^{6,7}, these new results suggest a dynamic switch in the source and type of CSF-1R ligands, where CSF-1 provided by neural progenitors and to a lesser extent by postmitotic neurons plays a dominant role during embryogenesis, while *Il34* increasingly contributes to microglial maintenance during the early postnatal period. We have included these results in the main manuscript and figures and clarified and expanded their discussion in the revised manuscript accordingly.

Clarity of Data Interpretation:

*In Figure 4, microglial repopulation in the *Emx1cre; Csf1fl/fl* model is discussed as evidence of transient CSF-1 dependence. However, the underlying mechanisms of this repopulation (proliferation vs. migration) are not well defined. Additional experiments, such as fate-mapping of remaining microglia, would strengthen the conclusions.*

While we agree that understanding the specific contribution of proliferation vs migration to microglial repopulation in *Emx1^{cre}; Csf1^{fl/fl}* model is of significant interest, we believe that a detailed dissection of these mechanisms falls beyond the scope of the present study.

Nevertheless, to offer deeper insights into how microglial repopulation may occur in this model, we have now included a more detailed characterization of microglial distribution across different subregions of the neocortex at P0, P3 and P7 in control and *Emx1^{cre}; Csf1^{fl/fl}* mutants (**Appendix Figure S3A-C**). While overall microglia density was lower in *Emx1^{cre}; Csf1^{fl/fl}* animals as compared to controls during the repopulation process, microglia distribution across the cortical layers remained similar to the one of controls, albeit layer IV was not fully repopulated by P7 in mutants. This suggests that neural-derived *Csf1* is not essential for establishing the overall laminar organization of microglia in the cortex. These new data offer a more nuanced understanding of the timing and dynamics of microglial repopulation and provide a solid framework for future studies.

Minor Comments:

Writing and Organization:

The manuscript is generally well-written but could benefit from streamlining in the results section to improve clarity. The discussion occasionally reiterates points already made in the results. Condensing this section while maintaining key insights would improve readability. Some claims, such as the generalization of findings to human microglia, should be softened or supported by additional evidence. The discussion should clearly delineate which findings are specific to mice and which might be extrapolated.

We thank the reviewer for these helpful suggestions. Accordingly, we have revised the Results section to streamline the presentation of our findings and improve clarity. Specifically, we have reduced redundancy, restructured certain paragraphs for better flow, and ensured that each sub-section focuses on a clear and distinct message.

Finally, we acknowledge the importance of clearly distinguishing mouse-specific findings from those that may be relevant to humans. In the revised manuscript, we have added data showing the expression of *Csf1* and *I134* (Figure EV2A-F), as detailed above, and explicitly indicated that our conclusions are based on murine data.

The manuscript states that both male and female mice were used in the experiments. However, the reporting of sex differences is inconsistent. While the text explicitly mentions that no differences in microglial proliferation or density were observed between males and females in some experiments, it is unclear whether all experiments were conducted with a balanced representation of both sexes or whether some used only a mixed cohort without sex-specific analysis.

To improve clarity, the authors should:

Clearly specify the sex of animals used for each experiment in the Methods section and figure legends.

Indicate whether data from males and females were pooled or analyzed separately. If pooled, justify why no sex-based differences were considered.

Explicitly state when statistical analyses for sex differences were performed, and if they were not conducted, provide a rationale.

We thank the reviewer for raising this important point. We have examined both males and females throughout our studies. To limit the number of animals, we have not systematically performed statistical tests comparing males and females, since for critical experiments, such as the density throughout development or the reduction of microglial density in *Csf1* conditional mutants, we did not observe statistically different distributions or phenotypes (microglial density across females and males assessed through Two-way ANOVA: cortex: $P=0.7916$; striatum: $P=0.8757$; POA: $P=0.5840$; hippocampus: $P=0.3631$ and microglial density in *Csf1* conditional *Cx3cr1creERT2* mutants across females and males assessed through unpaired Mann-Whitney tests $P=0.70$). We therefore pooled males and females in our analyses, but have made available the information related to the sexual identity of embryos and pups analyzed in the resource files for data presented in main figures and in the legends of supplemental figures, for most of the analyses. This enables the readers to find the information and assess the datasets. To fully clarify our rationale and findings, we have added a paragraph in the material and methods, detailing our analyses and the availability of information regarding sexual identity. We believe that this brings important clarification to our study as well as transparency and data availability for the readers.

Figures and Data Presentation:

Some figure panels (e.g., Figure 1G-J) are dense and difficult to interpret. The use of additional annotations or simplified schematics would help readers grasp key findings more easily.

We thank the reviewer for this suggestion, and have now revised Figure 1G-J, using simplified schematics to better guide the reader through key findings and to improve clarity.

Figure S7 should be integrated into the main figures to highlight the ATM-specific expression of CSF-1 more prominently.

We agree that the ATM-specific expression of *Csf1* is a critical point. As suggested, we have performed additional and new experiments in this area and these data are now shown as Figure 6A.

Reproducibility and Methods:

The methods section is thorough, but some aspects (e.g., quantification strategy for RNAscope) should be more explicitly detailed to ensure reproducibility. The authors mention statistical analyses but should provide effect sizes for key comparisons to better contextualize their findings.

We thank the reviewer for these helpful suggestions. To improve transparency and reproducibility, we have now revised the Methods section to provide a more detailed description of the quantification strategy used for RNAscope analysis, including criteria for cell segmentation.

Conclusion:

This study provides significant insights into the developmental dynamics of microglial colonization and proliferation, particularly the role of CSF-1 from distinct cellular sources. The findings are compelling and will be of broad interest to neurodevelopmental and neuroimmunology researchers. However, additional experimental validation and improved clarity in data interpretation are needed to fully substantiate the claims. Addressing these points will strengthen the manuscript and enhance its impact.

We are grateful to the reviewer for acknowledging the interest and thoroughness of our study. In the revised version of our manuscript, we have added novel experimental validation of our findings and improved the clarity of some results so that their substantiation of our initial claim is stronger and more clear. We believe that with these additional experiments and analyses address the reviewer's comments and significantly improve the strength of our manuscript.

Referee #2:

In this manuscript, Bridlance et al. aim to show how microglial colonization of the developing mouse brain occurs. The authors provide data that suggest microglial proliferation requires both extrinsic growth factor from neurons and intrinsic growth factor from axon tract-associated microglia (ATMs), but does not require normal neuronal activity. Given the importance of microglia as innate immune cells of the brain that regulate neural networks and are central to brain health and disease, understanding how microglia develop properly is important and of widespread interest. In this report, the authors use Cx3cr1gfp/+ mice and perform brain immunohistochemistry with proliferative markers at successive developmental time points with beautiful microscopy images to characterize microglial proliferation over time in the developing brain. Then, using Cre-lox genetic recombination system in Emx1Cre/+ ; Kir2.1 (floxed stop), Emx1Cre/+ ; Csf1fl/fl , and Cx3cr1Cre/+ ; Csf1fl/fl mice the authors interrogate the roles of neural activity, neural Csf1, and microglial Csf1 on microglial colonization. Lastly, the authors perform scRNAseq and RNA in situ hybridization to further characterize how developing microglial transcriptomes change without Csf1 from Emx1-expressing cells.

We are grateful to the reviewer for acknowledging the strength, interest and novelty of our findings and for providing constructive feedback.

While these studies are well-conducted and of potentially broad interest to the neuroscience community, there are two major concerns that if addressed would greatly improve the impact of this work

1) Neuronal activity is not required for microglial colonization. Using Emx1Cre/+ ; Kir2.1 (floxed stop) mice, the authors find no change in microglial colonization in response to presumed hyperpolarization of the circuit. They follow with whisker plucking or infraorbital nerve injury to show no changes in colonization. Given that these sensory deprivation experiments can lead to increased pruning and thus provide a theoretical counterbalance to any potential depletion, it becomes important that the authors prove hyperpolarization in their circuit with the Emx1Cre+ model by citation or physiology. In addition, while density was unchanged there is certainly a different characteristic noted in the Figure 2G particularly throughout which calls into question the observation based on density. Perhaps absolute number count or some higher power imaging can help adjudicate the Emx1 Cre Kir experiments and strengthen the argument.

We thank the reviewer for this important point. To strengthen the interpretation of our *Emx1^{Cre/+}; Rosa^{Kir2.1}* experiment, we have added a citation demonstrating that this model has been well-characterized and shown to induce effective neuronal hyperpolarization and drastic activity changes in the targeted cortical circuits⁸.

Regarding microglial density, we agree with the reviewer that density alone might not fully capture changes in microglial colonization or morphology. While we were largely consistent with our density measures, we have now added absolute microglial cell counts for all the experiments where neuronal activity was altered (**Appendix Figure S2A-2C**). These data confirm that, despite altered neuronal activity, microglial number and overall colonization are not significantly changed, supporting our conclusion.

Our findings support a model in which microglial colonization of the developing brain occurs in at least two distinct phases. During the early embryonic period, microglial proliferation and migration rely predominantly on CSF-1 signaling from neural progenitors. This phase ensures the initial seeding and expansion of microglia throughout the brain parenchyma,

largely independent of neuronal activity. Indeed, genetic depletion of CSF-1 in progenitors leads to drastic reductions in microglial number during embryogenesis, highlighting the critical trophic role of this cytokine. Following this CSF-1-dependent colonization phase, our data and a recent study suggest a subsequent developmental window in which microglial proliferation and distribution become increasingly influenced by neuronal activity⁹. Postnatally, as neural circuits mature and neuronal firing patterns emerge, microglia adapt their density and morphology in response to local activity cues, contributing to synaptic pruning and circuit refinement. This shift in microglial regulation from a growth-factor-driven to an activity-dependent mechanism likely reflects the changing demands of the maturing brain environment.

Together, these phases underscore the dynamic and context-dependent regulation of microglia during brain development, with CSF-1 guiding early expansion and neuronal activity shaping later microglial density, maturation and function.

2) Neuronal Csf1. The authors conclude that Csf1 expression "later in astrocytes and oligodendrocytes" was targeted using their Emx1Cre/+ ; Csf1fl/fl mice; there are no validation experiments demonstrating specificity of Csf1 reduction. Because the authors conclude neural Csf1 is required for microglial proliferation, the authors need to show validation of their knockout. Specifically, the authors could perform RNA in situ for Csf1 transcripts along with astrocyte, neuron, and oligodendrocyte markers to quantify the impact of Csf1 depletion of microglial colonization or clarify that it is parenchymal extrinsic Csf1 rather than neuronal.

We agree with the reviewer that validating the specificity of *Csf1* deletion in our *Emx1^{cre}; Csf1^{fl/fl}* model is important. Accurately measuring *Csf1* expression is technically challenging due to the existence of three isoforms, including a secreted form. We have attempted detection through Western blot and immunostaining, but these approaches do not fully capture the complexity or spatial resolution required to assess isoform- and cell type-specific expression. Nevertheless, using RNAscope experiments, we now show the specific local depletion of *Csf1* within the neocortex, which has been included in a new figure panel (Figure EV2H).

We also wish to point out that astrocytes primarily emerge during early postnatal life, and are largely absent during embryogenesis. While oligodendrocyte progenitor cells (OPCs) are present at embryonic stage, they arise from an *Emx1*-negative population and thus, are not affected in our model¹⁰. Moreover, based on several mouse and human transcriptomic datasets, *Csf1* appears to be predominantly expressed by neural progenitors, and to a lesser extent by post-mitotic neurons, before the later emergence of astrocytes and oligodendrocytes.

Taken together, although technical limitations prevent precise validation of *Csf1* deletion at the single-cell level, developmental timing and expression data support the idea that the relevant source of *Csf1* during embryogenesis is neural parenchymal but non-glial—likely progenitors and early neurons. Therefore, we have revised the text to clarify that our findings point to a requirement for neural extrinsic *Csf1*, rather than strictly neuronal *Csf1*, in supporting microglial proliferation during development.

Minor concerns.

1) Morphological changes. In the text of the manuscript, the authors refer to amoeboid and ramified morphologies in developmental hotspots. To strengthen these claims, the authors should include quantification of these morphologies: number or length of processes,

individual fractal or skeletal analysis, or cell body area/perimeter analysis. For example: PMID: 36307475

We agree that morphological quantification is a valuable tool to characterize some microglial states and thank the reviewer for highlighting the need for additional clarity. During development, microglia exhibit highly dynamic and heterogeneous morphologies, transitioning rapidly between amoeboid, ramified, and intermediate forms. This dynamic nature is well captured in our live imaging movies, which highlight the transient and fluid morphological changes rather than static states (Figure EV1). Consequently, traditional static morphological metrics such as process length or fractal analysis may only provide a limited snapshot that does not fully represent microglial dynamics during these critical developmental windows. We have clarified this point in the revised manuscript to explain that we acknowledge the diversity of microglial forms observed. These results are also consistent with previous work describing the amoeboid microglial morphology found at the CSA¹¹ and EDWM¹²⁻¹⁵.

2) In panels 1L and 1N, the graphs do not extend to the P14 timepoint even though the text body suggests that at P14 proliferation becomes undetectable. These data should be included.

We thank the reviewer for pointing out this lack of clarity. We have updated the text to clarify that both accumulations are no longer observed by P14, as illustrated in Figures 1F and 1J. Thus, we cannot assess proliferation at this stage, which is why these data were not included in the graphs. This has been explicitly stated in the revised manuscript to ensure clarity.

3) Lastly, the conclusions of this paper suggest that there is a dose-dependent effect but also a cell-type specific effect of CSF1 production - there is no evidence to suggest the dose dependent effect. The authors could quantify Csf1 expression in their models as suggested above or overexpress CSF1 in a non-neuronal cell type or even simply inject more CSF1 in the Emx1Cre/+ ; Csf1fl/fl mice.

We thank the reviewer for raising this important point. As noted previously, quantifying *Csf1* levels is technically challenging due to the presence of multiple isoforms (including a secreted form) and limitations in currently available tools. We attempted several approaches, including Western blot and immunostaining, but these methods did not provide reliable or interpretable quantification of *Csf1* across cell types or conditions.

While we could not directly measure *Csf1* levels, we do observe a dose-dependent effect on microglial numbers across different models. For instance, the *Emx1^{cre}; Csf1^{fl/+}* heterozygotes show an intermediate reduction in microglial numbers compared to *Emx1^{cre}; Csf1^{fl/fl}* homozygotes, suggesting that the extent of *Csf1* depletion correlates with microglial proliferation or maintenance in the brain. This phenotypic gradient supports the idea of a functional dose-dependency, even if *Csf1* itself cannot be directly quantified.

We have clarified this point in the revised manuscript and now refer to a dose-dependent effect on microglia, rather than a demonstrated dose-dependent regulation of *Csf1* expression *per se*.

4) Cre controls - in knock-in knockout systems, assessing the impact of Cre is important in models. Suggest the authors in the future include Cre only controls in their analyses to strengthen their conclusions. PMCID: PMC5910175

We thank the reviewer for this important suggestion. We have included Cre-positive, floxed-negative controls in our analyses and confirmed that Cre expression alone does not impact the parameters measured. This has been added to the methods and results (**Figure EV3D-F**) to strengthen the conclusions.

Referee #3:

Bridance et al. demonstrate that microglia undergo two distinct waves of proliferation during forebrain development in mice, gathering transiently in specific hotspots where they adopt a proliferative Axon-Tract-associated Microglia (ATM)-like phenotype. The authors convincingly show that prenatal and early postnatal colonization patterns of microglia are largely independent of neuronal activity. Using several transgenic mouse models with conditional CSF1 inactivation, the authors provide clear evidence that embryonic cortical microglia critically depend on locally produced neural-derived CSF1, predominantly originating from cortical progenitor cells and, to a lesser extent, post-mitotic neurons. This action of CSF1 is local, dose-dependent, and transient. Notably, intrinsic CSF1 expression by ATM itself contributes to sustaining their proliferation within developmental hotspots. The paper is well-written, structured logically, and significantly clarifies the dynamics of microglial colonization in the developing brain. Furthermore, it opens new avenues for investigating the region- and stage-specific roles of microglia during embryonic brain development.

Novelty: While the reliance of microglia on CSF1 for proliferation and survival has been previously documented, including the role of neuronal-derived CSF1 in microglial number colonisation regulation in late embryonic and postnatal cerebellar development, this study uniquely illustrates the critical role of neural-derived CSF1 in controlling microglial colonization and proliferation specifically in the developing neocortex. Additionally, the authors confirm the role for ATM microglia in producing intrinsic CSF1, and demonstrate it further promotes their own proliferation.

We are grateful to the reviewer for acknowledging the strength, interest and novelty of our findings and for providing constructive feedback.

Major Comments:

1. It would be valuable for the authors to clarify if microglial colonization in the neocortex depends on a CSF1 gradient. Does the local CSF1 gradient influence microglial migration in addition to proliferation?

We thank the reviewer for this insightful question. While the potential role of a local CSF-1 gradient in directing microglial migration is of significant interest, we believe that this aspect is beyond the scope of the current study.

Nevertheless, to offer deeper insights into how microglial repopulation may occur in this model, we have now included a more detailed characterization of microglial distribution across different subregions of the neocortex at P0, P3 and P7 in control and *Emx1^{cre}; Csf1^{fl/fl}* mutants (**Appendix Figure S3A-C**). While overall microglia density was lower in *Emx1^{cre}; Csf1^{fl/fl}* animals as compared to controls during the repopulation process, microglia distribution across the cortical layers remained similar to the one of controls, albeit layer IV was not fully repopulated by P7 in mutants. This suggests that neural-derived *Csf1* is not essential for establishing the overall laminar organization of microglia in the cortex. These new data offer a more nuanced understanding of the timing and dynamics of microglial repopulation and provide a solid framework for future studies.

2. Given prior evidence of CSF1R expression on neuronal progenitors and cortical neurons, it is important to address if the presented CSF1 effect is direct or indirect. Does neuronal-CSF1 stimulate neuronal populations that might in turn influence microglial dynamics?

We thank the reviewer for this insightful question. The potential for CSF1 to act directly on neuronal or progenitor populations via CSF1R is an important consideration.

To address this, we first showed that *Csf1r* expression is not detected in mouse or human neural progenitors and post-mitotic neurons during prenatal development (**Figure EV2D-E and G**). Second, we performed RNAscope analysis at E14.5 to assess the cellular localization of *Csf1r* mRNA in the developing neocortex. Our data revealed that *Csf1r* expression is restricted to microglial cells (**Figure EV2G**). These findings indicate that the prenatal effects of neural-derived *Csf1* on microglial dynamics are unlikely to be mediated indirectly through CSF1R-expressing neural populations. Instead, they support the conclusion that, during the embryonic period relevant to our study, *Csf1* acts primarily through microglial CSF1R, rather than through a direct effect on neuronal or progenitor populations.

*3. Could the authors assess if there are other (neuronal or glial) changes beyond microglial numbers in the *Emx1cre/+;Csf1fl/+* and *Emx1cre/+;Csf1fl/fl* models? Performing transcriptomic analyses on neuronal and glial populations or bulk would greatly enhance understanding and help address potential indirect effects.*

As part of our study, we performed single-cell RNA sequencing at P3, and these data are presented in the manuscript (**Figure 4F-L**). This analysis includes major neuronal, glial, and microglial populations, allowing us to detect potential transcriptomic alterations beyond microglial changes in the *Emx1^{Cre}; Csf1^{fl/fl}* animals. That said, we acknowledge the challenge of distinguishing direct effects of CSF-1 on neural populations from indirect effects resulting from microglial depletion.

A comprehensive characterization of how the absence of microglia affects other brain cell populations is an important line of investigation, but is somewhat beyond the scope of the current study, which is centered on identifying the source and developmental timing of CSF-1 required for microglial proliferation.

REFERENCES

- 1 Javed, A. *et al.* Developmental gene expression patterns driving species-specific cortical features. *bioRxiv*, 2025.2002.2018.638637, doi:10.1101/2025.02.18.638637 (2025).
- 2 Baldassari, S. *et al.* Single-cell genotyping and transcriptomic profiling of mosaic focal cortical dysplasia. *Nat Neurosci* **28**, 964-972, doi:10.1038/s41593-025-01936-z (2025).
- 3 Velmeshev, D. *et al.* Single-cell analysis of prenatal and postnatal human cortical development. *Science* **382**, eadf0834, doi:10.1126/science.adf0834 (2023).
- 4 Harris, S. E. *et al.* Meox2Cre-mediated disruption of CSF-1 leads to osteopetrosis and osteocyte defects. *Bone* **50**, 42-53, doi:10.1016/j.bone.2011.09.038 (2012).
- 5 Faust, T. E. *et al.* A comparative analysis of microglial inducible Cre lines. *Cell Rep* **42**, 113031, doi:10.1016/j.celrep.2023.113031 (2023).
- 6 Greter, M. *et al.* Stroma-derived interleukin-34 controls the development and maintenance of langerhans cells and the maintenance of microglia. *Immunity* **37**, 1050-1060, doi:10.1016/j.immuni.2012.11.001 (2012).
- 7 Devlin, B. A. *et al.* Excitatory-neuron-derived interleukin-34 supports cortical developmental microglia function. *Immunity*, doi:10.1016/j.immuni.2025.06.002 (2025).
- 8 Anton-Bolanos, N. *et al.* Prenatal activity from thalamic neurons governs the emergence of functional cortical maps in mice. *Science* **364**, 987-990, doi:10.1126/science.aav7617 (2019).
- 9 Kumaraguru, S., Morgan, J. & Wong, F. K. Activity-dependent regulation of microglia numbers by pyramidal cells during development shape cortical functions. *Sci Adv* **11**, eadq5842, doi:10.1126/sciadv.adq5842 (2025).
- 10 Kessaris, N. *et al.* Competing waves of oligodendrocytes in the forebrain and postnatal elimination of an embryonic lineage. *Nat Neurosci* **9**, 173-179, doi:10.1038/nn1620 (2006).
- 11 Lawrence, A. R. *et al.* Microglia maintain structural integrity during fetal brain morphogenesis. *Cell* **187**, 962-980 e919, doi:10.1016/j.cell.2024.01.012 (2024).
- 12 Nemes-Baran, A. D., White, D. R. & DeSilva, T. M. Fractalkine-Dependent Microglial Pruning of Viable Oligodendrocyte Progenitor Cells Regulates Myelination. *Cell Rep* **32**, 108047, doi:10.1016/j.celrep.2020.108047 (2020).
- 13 Li, Q. *et al.* Developmental Heterogeneity of Microglia and Brain Myeloid Cells Revealed by Deep Single-Cell RNA Sequencing. *Neuron* **101**, 207-223 e210, doi:10.1016/j.neuron.2018.12.006 (2019).
- 14 McNamara, N. B. *et al.* Microglia regulate central nervous system myelin growth and integrity. *Nature* **613**, 120-129, doi:10.1038/s41586-022-05534-y (2023).
- 15 Hammond, T. R. *et al.* Single-Cell RNA Sequencing of Microglia throughout the Mouse Lifespan and in the Injured Brain Reveals Complex Cell-State Changes. *Immunity* **50**, 253-271 e256, doi:10.1016/j.immuni.2018.11.004 (2019).

Dear Morgane,

Thank you again for submitting your revised manuscript (EMBOJ-2025-120269R) to The EMBO Journal for our consideration, and for your patience during peer review. Your manuscript has been sent back to the three original referees that had previously reviewed the initial version of your manuscript, and we have now received their comments, which you can find below.

I am very pleased to say that all three referees are satisfied with the revision, mention that all previously raised concerns have been thoroughly and sufficiently addressed in the strengthened new version of the manuscript, and now support its publication without any further comments. In light of this input, I am glad to let you know that your manuscript has been accepted in principle for publication in The EMBO Journal.

Before we can move forward with formal acceptance and publication, there are a few changes from the editorial side we need you to make in a final version:

- Please note that all relevant funding information provided in the Acknowledgments section of the manuscript should also be entered in our online manuscript handling system; currently, the following information is missing from the online system: "AMX PhD fellowship and the Fondation ARC pour la recherche sur le cancer (ARCD0C42022010004628); the Fondation pour la Recherche Médicale (FRM, EQU202003010195)".
- Please remove the "Lead contact" information from the title page (only the contact information of the co-corresponding authors should remain) and from the Methods section; "Resource availability", "Lead contact", "Materials availability", and "Data and code availability" sections in the Methods should be merged into a single new section named "Data availability".
- Please change heading "Material and Methods" to "Methods", and include the subheading "Methods and Protocols" instead of headings "Experimental models" and "Method details". For more information on our structured methods format, please see our guide to authors: <https://www.embopress.org/page/journal/14602075/authorguide#structuredmethods>.
- Please make sure that the RNA-seq data reported in your manuscript will be publicly available at the time of publication from the ArrayExpress database, and provide in the "Data availability" section the permanent and specific URL to the dataset.
- Please change heading "Declaration of interests" to "Disclosure and competing interests statement".
- The author contributions statement should be removed from the manuscript file. Instead, we use CRediT to specify the contributions of each author in the journal submission system. Please feel free to use the free text box to provide more detailed descriptions during submission. See also our guide to authors for more information: <https://www.embopress.org/page/journal/14602075/authorguide#authorshippinguidelines>.
- Please note that all Figure panel callouts should be listed sequentially.
- We also noticed that callouts for Fig. 4L are missing.
- Please state in the "Mouse lines" section of your Methods details of the authority granting ethics approval, including the reference number for approval, for your experiments involving animals.
- Appendix Figures should only be included in the Appendix PDF file, not uploaded individually.
- "Table S1" and the corresponding callouts should be renamed to "Table EV1"; its legend must be included above the table in the Excel file.
- Please note that EMBO press papers are accompanied online by:
 - A) a short (2 sentences) summary of the findings and their significance,
 - B) 2-5 short bullet points highlighting the key results, and
 - C) a synopsis image in .jpg or .png format that is exactly 550 pixels wide and 300-600 pixels high (the height is variable). Please note that all text in the image needs to be legible at the final size.Please upload this information along with your revised manuscript (the text for A and B should be provided in a separate Word file).
- During our routine pre-acceptance Figure checks, our Data Integrity analyst detected possible reuse between Figure 1B (P7 Neocortex) and Figure 2A (P7). Please check these Figures again and let us know whether this reuse is intentional. In this case, please clearly state the reuse in the respective Figure legends. Otherwise, please revise the Figures to avoid reuse.
- During our routine data checks, our data editors have raised the following queries/concerns regarding data shown in Figures,

statistics, and legends. Please make sure that all requests below are completely addressed (and all changes highlighted) in the final version of the manuscript:

1. Please provide the exact p-values in the legends of Figures 3F, H, J, M; 4B, D, E; 5G, J; 6C, E, G, H, J, K; EV1 F, EV3 I, EV5 J-L; S3B, C.
2. Please indicate the statistical test used for data analysis in the legend of Figure 4G.
3. Please note that information related to "n" is missing in the legends of Figures 2D, 4G.
4. Please note that the error bars are not defined in the legends of Figures 1C, E.
5. Please note that the scale bar needs to be defined for Figures EV2 G, H.
6. Please note that the asterisk is not defined in the legend of Figure EV1 B. This needs to be rectified.
7. Please note that the arrow heads are not defined in the legend of Figure 1B. This needs to be rectified.
8. Please note that the closed arrow heads are not defined in the legend of Figure 4H. This needs to be rectified.

- The order of the manuscript sections must be corrected as follows: Title page - Abstract and Keywords - Introduction - Results - Discussion - Methods - Data Availability - Acknowledgements - Disclosure and Competing Interests Statement - References - Figure Legends - (main Tables with legends, if applicable) - Expanded View Figure Legends.

Please also note that as part of the EMBO publications' Transparent Editorial Process, The EMBO Journal publishes online a Peer Review File along with each accepted manuscript. This File will be published in conjunction with your paper and will include the referee reports, your point-by-point response and all pertinent correspondence relating to the manuscript. You can opt out of this by letting the editorial office know (contact@embojournal.org). If you do opt out, the Peer Review File link will point to the following statement: "No Peer Review File is available with this article, as the authors have chosen not to make the review process public in this case."

We look forward to seeing a final version of your manuscript as soon as possible. Please let us know if you have any questions and use this link to submit your revision: <https://emboj.msubmit.net/cgi-bin/main.plex>.

Best regards,

Ioannis

Referee #1:

This revised manuscript offers a comprehensive and well-supported analysis of the dual roles of extrinsic and intrinsic CSF-1 in regulating microglial colonization during early brain development. The authors provide robust experimental evidence using complementary genetic models, RNA analyses, and imaging. All reviewer comments have been thoroughly addressed through additional experiments or revisions. The paper is suitable for publication in its current form.

Referee #2:

The authors have more than adequately addressed all of this reviewer's concerns.

Referee #3:

The Authors sufficiently responded to the concerns.

All editorial and formatting issues were resolved by the authors.

Dear Morgane,

Congratulations on an excellent manuscript! I am very pleased to inform you that it has been accepted for publication in The EMBO Journal. Thank you for comprehensively addressing the initially raised referee criticisms and the editorial requests for corrections and changes.

If you have any questions, please do not hesitate to contact the Editorial Office. Thank you for your contribution to The EMBO Journal. Working with you has been a pleasure!

Best regards,

Ioannis
